# Emergence of a geometric pattern of cell fates from tissue-scale mechanics in the *Drosophila* eye

Kevin D Gallagher[1,2], Madhav Mani[1,2,3]*, Richard W Carthew[1,2]*

[1]Department of Molecular Biosciences, Northwestern University, Evanston, United States; [2]NSF Simons Center for Quantitative Biology, Northwestern University, Evanston, United States; [3]Department of Engineering Sciences and Applied Mathematics, Northwestern University, Evanston, United States

**Abstract** Pattern formation of biological structures involves the arrangement of different types of cells in an ordered spatial configuration. In this study, we investigate the mechanism of patterning the *Drosophila* eye epithelium into a precise triangular grid of photoreceptor clusters called ommatidia. Previous studies had led to a long-standing biochemical model whereby a reaction-diffusion process is templated by recently formed ommatidia to propagate a molecular prepattern across the eye. Here, we find that the templating mechanism is instead, mechanochemical in origin; newly born columns of differentiating ommatidia serve as a template to spatially pattern flows that move epithelial cells into position to form each new column of ommatidia. Cell flow is generated by a source and sink, corresponding to narrow zones of cell dilation and contraction respectively, that straddle the growing wavefront of ommatidia. The newly formed lattice grid of ommatidia cells are immobile, deflecting, and focusing the flow of other cells. Thus, the self-organization of a regular pattern of cell fates in an epithelium is mechanically driven.

*For correspondence:
madhav.mani@gmail.com (MM);
r-carthew@northwestern.edu
(RWC)

**Competing interest:** The authors declare that no competing interests exist.

## Editor's evaluation

This paper will have a high impact for biologists who are interested in tissue patterning and organogenesis. It provides unexpected insights into the problem of regular spacing of sub-organ structures. The study is based on innovative live imaging technology with state of the art analysis tools.

## Introduction

The spatial arrangement of distinct cell types in organs can be highly precise. Such regularity in spacing is observed in a diversity of systems, such as hairs in mammals and feathers in birds, and within a variety of sensory organs (*Schweisguth and Corson, 2019*). Often, the regularly spaced array comprises groups of cells with a common function that ensures a regular field of functionality in an organ. For example, the grid cells of the mammalian cortex fire in a triangular grid pattern to encode a neural representation of Euclidean space (*Rowland et al., 2016*). Likewise, the photoreceptor cells (R cells) of the insect compound eye fire in a triangular grid pattern to sense the visual environment with great acuity (*Nilsson, 1989*). Manifestly, all of these highly regular body structures emerge over time from more disordered epithelial and mesenchymal tissues. Yet, we lack an understanding of the principles at play in the emergence of order from disorder in living systems.

Previous studies have proposed that to form an ordered spacing of functional units, a regular prepattern of differential gene expression initially arises, followed by cell differentiation and morphogenesis (*Hiscock and Megason, 2015*; *Schweisguth and Corson, 2019*). The prepattern is thought to

originate from smoothly varying levels of key molecular factors and cell-cell interactions to generate a spatially regular pattern of gene expression. A particular example of this class of models, the reaction-diffusion model, posits that a molecular activator and inhibitor react with one another and effectively diffuse out from their sources of synthesis, from which emerges a regular pattern of cell types (*Meinhardt, 1982*; *Oster and Murray, 1989*; *Turing, 1952*). Mechanical reaction-diffusion mechanisms have also been proposed, where the activators and inhibitors are not molecules but mesenchymal cells that can freely migrate to form spaced patterns (*Glover et al., 2017*; *Painter et al., 2018*; *Shyer et al., 2017*; *Volkening, 2020*).

Similar to the engineering of crystals in industrial contexts, spaced patterns in living systems often progressively emerge in a directional manner, initializing at one region and then propagating across a tissue (*Sato et al., 2013*; *Schweisguth and Corson, 2019*). The rationale for such a sequential mechanism is to ensure coherent ordering across the entire lattice, absent any defects, since the propagating wavefront of pattern formation acts as a template for the addition of new units. Evidence to support this rationale comes from avian feather follicle development, where the precise arrangement of follicles in the chick is impaired if the wavefront is decoupled from follicle self-organization (*Ho et al., 2019*). Bird species that naturally lack a precise arrangement of feather primordia develop without a wavefront (*Ho et al., 2019*). In the *Drosophila* retina, if two wavefronts are triggered then errors in R cell spacing occur as a result (*Heberlein et al., 1995*; *Ma and Moses, 1995*).

The *Drosophila* retina has been a well-studied model of how an ordered lattice of functional units can emerge (*Ready et al., 1976*). There is a wavefront of gene expression that initializes at the posterior edge of the epithelium and propagates across the field over a two-day period (*Roignant and Treisman, 2009*). Cells are thought to remain motionless within the plane of the epithelium (*Ready et al., 1976*). Hence, they experience the wavefront of gene expression that is then passed along to their anterior neighbors. The gene *atonal* is an essential member of the wave, expressing a proneural transcription activator (*Jarman et al., 1994*). Atonal activates expression of the *scabrous* gene (*Jarman et al., 1994*), which encodes a secreted glycoprotein (*Baker et al., 1990*; *Lee et al., 1996*). Scabrous is critical to maintain the regular spacing of ommatidia – the clusters of R cells that form a functional unit eye. Loss of Scabrous results in ommatidia that are spaced too close to one another and exhibit loss of regularity (*Baker et al., 1990*; *Wolff and Ready, 1991*). Atonal expression also becomes crowded and irregular in *scabrous* mutants (*Lee et al., 1996*). Reaction-diffusion models have proposed Scabrous as a long-range inhibitor of the activator Atonal, and such models successfully simulate the sequential emergence of ordered ommatidia in a tissue of stationary cells (*Lubensky et al., 2011*).

We began to question this model because it failed to explain a conserved feature of compound eye development found in *Drosophila* and other insect species (*Melamed and Trujillo-Cenóz, 1975*; *Ready et al., 1976*). The wavefront of gene expression is coincident with a physical deformation of the epithelium brought about by a constriction of cells' apical domains to form a dorsal-ventral furrow called the morphogenetic furrow (MF) (*Wolff and Ready, 1991*). As the wave passes, most cells expand their apical domains on the posterior side except for the clusters of nascent R cells, which remain apically constricted. Thus, the wave of gene expression is coupled to a mechanical wave (*Brown et al., 2006*; *Escudero et al., 2007*; *Wolff and Ready, 1991*). This is not a unique feature of the compound eye, and coupled mechanochemical waves have been observed at fronts of differentiation in a variety of tissues (*Brodland et al., 1994*; *Harris et al., 1984*; *Sato et al., 2013*). Reaction-diffusion models successfully mimic pattern formation in vivo without any obligatory cell shape or cell flow dynamics (*Gavish et al., 2016*; *Lubensky et al., 2011*), and myosin-induced apical constriction in the furrow is regulated by the signals that also regulate *atonal* expression (*Escudero et al., 2007*). For these reasons, the mechanical aspects of the phenomena have been considered to be surrogate to the dynamics of gene expression.

Here, we have explored the purpose of the mechanical wave in the retina by recording development via time-lapse microscopy. Epithelial cells within the wavefront experience a spatially periodic flow profile, moving cells in an anterior direction, with groups of cells flowing at the same velocity as the wavefront, alternated by groups of cells slowing down and being left behind the wavefront. These latter groups become a new column of ommatidia. The periodicity in flow along the wavefront is generated by the youngest three columns of ommatidia, which form a stationary lattice that deflects and focuses the anterior-wards flow of other cells around them. The source of the anterior

flow corresponds to a narrow zone of dilating cells positioned three columns behind the moving wavefront. Scabrous is required for these cells to fully dilate, and its loss results in a correlated diminishment of cell flows and irregularity in the flow periodicity.

## Results

The *Drosophila* retina develops from a larval eye imaginal disc that grows in size within the body of the larva. Approximately 36 hr before pupariation, the mechanochemical wave begins to move across the eye disc from the posterior edge (*Figure 1A*). Over the course of 2 days, the MF crosses the disc (*Ready et al., 1976*). As it migrates to the anterior, a column of regularly spaced ommatidia emerge every two hours (*Wolff and Ready, 1991*). Each emergent ommatidium is composed of five cells that are destined to become R2, R3, R4, R5, and R8 photoreceptors (*Figure 1B*). Cells within an ommatidium remain apically constricted and form a rosette structure while non-ommatidia cells are structurally disordered (*Figure 1B*). Subsequently, each ommatidium recruits cells to become R1, R6, and R7, whose apical faces undergo constriction. The regular spacing of ommatidia between two adjacent columns is out of phase such that a regular triangular grid, with odd and even columns of ommatidia, is generated in the domain posterior to the MF.

We developed a method to culture excised eye discs in vitro. Our technique involves surgically excising the eye-antennal disc complex from late third instar larvae, separating it from the brain, and culturing it in a dish (*Figure 1C*). The culture media was designed to mimic the composition of the circulatory hemolymph of larvae. Recent advancements made discovery of an optimal growth media feasible (*Dye et al., 2017*; *Tsao et al., 2016*), an endeavor that traces back to the development of Schneider's growth media (*Schneider, 1966*). Disc explants routinely sustained development for 10–12 hr, which enabled us to capture a rich set of developmental events from a single sample. Discs were scanned by fluorescence microscopy every 5 min over the time-course of disc culture. Fluorescence of GFP-tagged E-cadherin protein was imaged and processed, allowing us to detect individual cells at the plane of the adherens junction (*Figure 1—figure supplement 1A-C*). Each image scan was then semi-automatically segmented to identify all cells within the scan (*Figure 1—figure supplement 1D*). We used a tracking algorithm to map cells across scans, obtaining for each individual disc a cell-per-cell correspondence between developmental time points (*Figure 1—figure supplement 1E*). Since the number of segmented cells varied over time due to cell division, cell delamination, boundary effects, and segmentation errors, we manually corrected all of these events over the complete time-course (*Video 1*). We then manually assigned a state classifier to R cells in all mature ommatidia that were imaged. Cell state classification was possible because fate-determined cells can be unambiguously identified without cell-specific markers, based on their apical morphology and position relative to their neighbors (*Tomlinson and Ready, 1987*; *Wolff and Ready, 1991*). Since every cell had been tracked beginning to end of disc culture, these state values were propagated backward in time to identify those cells that were fated to become R cells (*Video 2*). In conclusion, we were able to follow the dynamics of eye development with single-cell resolution for 10 hr of time.

### Eye discs develop ex vivo as they do in vivo

Cultured eye discs exhibited many of the hallmarks of eye development described or inferred in previous studies. The MF progressed from posterior to anterior, and a new column of ommatidia appeared at intervals of 2.0–2.5 hr (*Figure 1D*). This rate of MF movement is in accordance with the estimated interval of two hours per column for in vivo development (*Campos-Ortega and Hofbauer, 1977*). Over the course of a typical imaging session, we were thus able to observe the emergence of 4–5 columns of ommatidia. The spacing of ommatidia in adjacent columns exhibited an alternating phase such that a triangular grid pattern of ommatidia emerged from the cultured eye discs (*Figure 1E*). Ommatidia emerged from the MF in stereotyped fashion. A line of adjacent cells at the posterior edge of the MF progressively bent to form multicellular arcs containing 10–15 cells (*Figure 1F*). These arcs evolved into immature rosettes of variable cell number, containing six to nine cells. This transition was driven by a series of intercalations that brought cells together in the dorsal-ventral direction and displaced cells to the anterior (*Figure 1—figure supplement 2A*). We observed that the cell fated to become R8 was typically at the most posterior position of the immature rosette (*Figure 1F*). The immature rosette evolved into a five-cell mature rosette, containing cells fated to differentiate into R2,

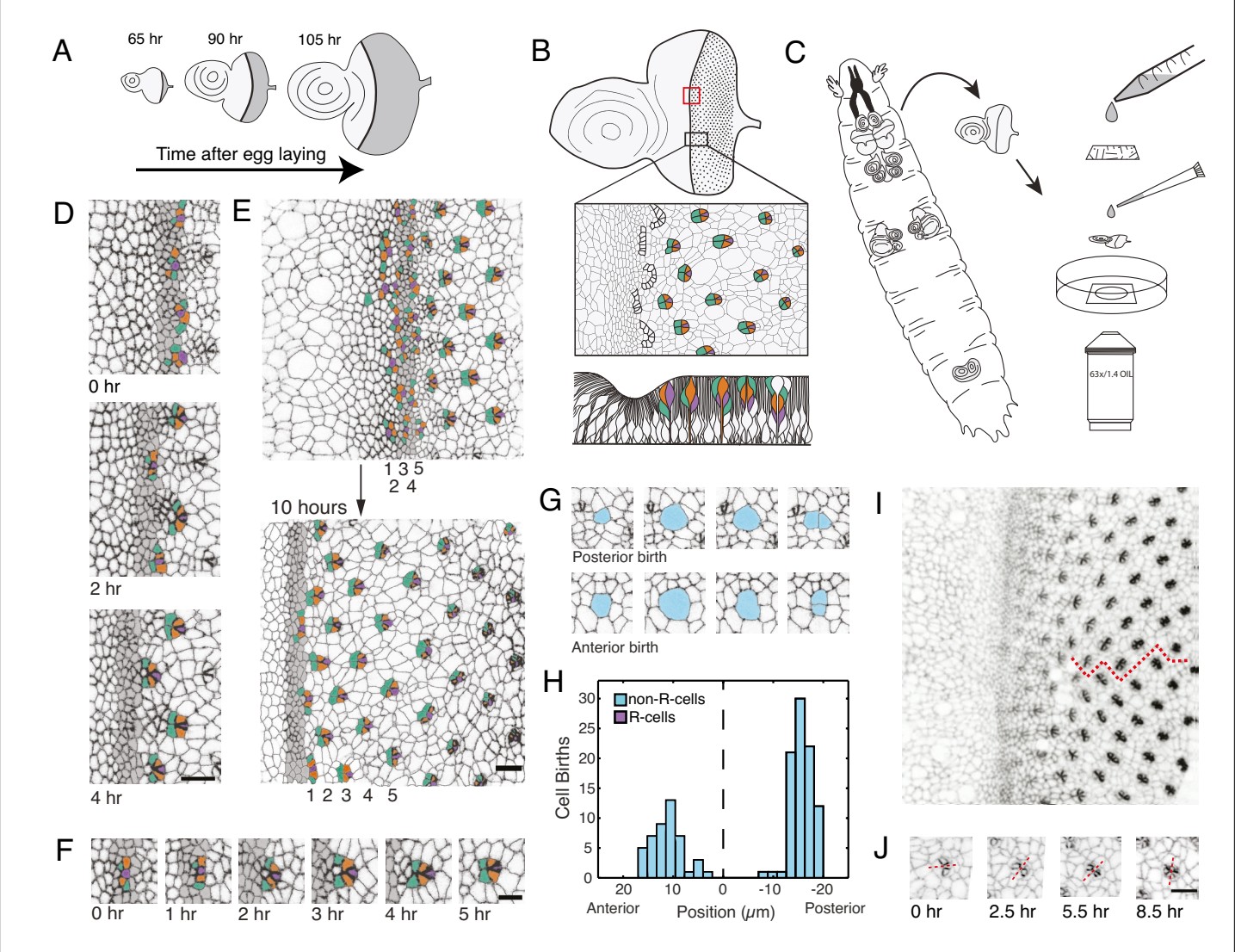

**Figure 1.** Living eye discs imaged by time-lapse microscopy. (**A**) Differentiation is initiated in the eye disc by the MF, which moves across the eye epithelium. As the MF transits the eye, cells on either side proliferate, which generates disc growth. Shown are three discs at various times after egg laying, the beginning of the animal's life. (**B**) Top, an apical view of an eye disc showing the triangular grid of ommatidia on the posterior side of the MF. A red rectangle outlines the region that was typically imaged by time-lapse microscopy; the black rectangle corresponds to the developmental sequence outlined below. Middle, camera lucida of apical plane of an eye disc. R cells are color coded: R8 (purple), R2 and R5 (orange), R3 and R4 (cyan). The other R cells are not colored but are differentiating in columns 4–6. Bottom, a cross-section view of the eye epithelium. Note the progenitor cell nuclei are basally positioned, and as they transition into a differentiated state, their nuclei migrate apically. Adapted after Figure 19 from *Wolff and Ready, 1993*. (**C**) Method of time-lapse microscopic imaging of an eye disc surgically excised from a late third-instar larva. (**D**) An E-cadherin GFP fusion protein outlines cell junctions, enabling cell segmentation and tracking. Two successive columns of ommatidia are formed at the indicated time points. R cells in this panel and all subsequent figure panels are colored as in panel B. (**E**) R cells constituting five columns of ommatidia (numbered 1–5) shown at the beginning and end of imaging. The ommatidia ultimately form a triangular lattice. (**F**) Time series for one cluster of presumptive R cells as they emerge from the MF and form a 5-cell rosette within 5 hours. The multicellular arc stage at 2–3 hr is followed by the immature rosette stage at 4 hr. (**G**) Cells located anterior and posterior to the MF undergo division. (**H**) Number of cell divisions recorded throughout the imaging session as a function of the cell's distance from the MF (dashed line). Note the division wave posterior to the MF is exclusively composed of non R-cells. (**I**) Frame from E-cadherin-GFP live imaging showing ommatidia at different stages of rotation with respect to the equator (red line). Rotation is clockwise for dorsal ommatidia and counter-clockwise for ventral ommatidia. (**J**) Successive time points of one ommatidium as it undergoes rotation. After rotating 45° by 2.5 hours, it resumes rotation at 5.5 hr and completes the 90° rotation after 8.5 hr. For panels D-J, anterior is to the left and dorsal to the top. Scale bar in panels D, E, and J is 5 µm. Scale bar in panel F is 3 µm.

The online version of this article includes the following figure supplement(s) for figure 1:

**Figure supplement 1.** Pipeline for image capture and processing.

**Figure supplement 2.** General features of ex vivo eye development.

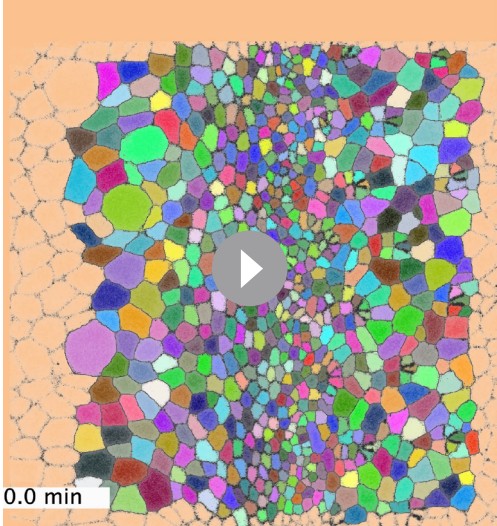

**Video 1.** A wildtype disc after segmentation and tracking.
Each unique tracked cell is labeled a specific color. Note that cells gain and lose color when they enter and exit the field of view, which marks the beginning and end, respectively, of those cells' tracks.
https://elifesciences.org/articles/72806/figures#video1

R3, R4, R5, and R8. One to three cells in the immature rosette, called mystery cells in prior studies (*Tomlinson and Ready, 1987*; *Wolff and Ready, 1991*), relaxed their apical domains and were ejected from the rosette (*Figure 1F*, *Figure 1—figure supplement 2A*). After establishment of the mature rosette, we observed recruitment of the other three photoreceptor types, R1, R6, and R7, as well as the recruitment of the non-neuronal cone cells (*Figure 1—figure supplement 2B*). The trajectory of observed ommatidium development was highly reproducible between different eye discs (*Figure 1—figure supplement 2C*) and entirely consistent with the trajectory inferred from fixed tissue experiments (*Escudero et al., 2007*; *Tomlinson and Ready, 1987*; *Wolff and Ready, 1991*).

Discs cultured ex vivo also recapitulated specific features of tissue growth inferred in vivo. Cell division was detected by the rounding up of cells on the adherens junction plane prior to cytokinesis, after which two daughter cells were observed (*Figure 1G*). Using this as a marker for cell division, we counted division events as a function of cell position along the anterior-posterior axis (*Figure 1H*). There was a zone of proliferation 10–20 µm anterior to the MF that abruptly ended at the anterior edge of MF. No cell division was detected in the MF and in a narrow region ~7 µm to its posterior. After this point, cells resumed proliferation but it was limited to a narrow zone of cells, and little proliferation was observed in more posterior regions. Proliferation in this zone was restricted to non-ommatidial cells, whereas cells in ommatidia remained non-proliferative (*Figure 1H*). There was no systematic pattern in proliferation detected along the dorsal-ventral axis. The complex pattern of proliferation we reproducibly observed mimics the in vivo cell division pattern (*Ready et al., 1976*; *Wolff and Ready, 1991*). Additionally, we observed a uniform distribution of cells that delaminated from the epithelium and disappeared from the apical surface of the tissue (*Figure 1—figure supplement 2D, D'*).

We also observed all the inferred morphological signatures of planar cell polarity (*Choi and Benzer, 1994*; *Ready et al., 1976*). Ommatidia in the fourth column posterior to the MF began to rotate relative to their cell neighbors (*Figure 1I*). Clusters rotated 90° and rotation was either clockwise or counterclockwise depending on which side of the equator (i.e. dorsal-ventral midline) the clusters were located. We observed that full rotation occurred in two steps of 45° each (*Figure 1J*).

Finally, we observed upregulated expression of E-cadherin protein in all cells within the MF, within the emerging rosettes, and within maturing ommatidia (*Figure 1I and J*). This upregulation

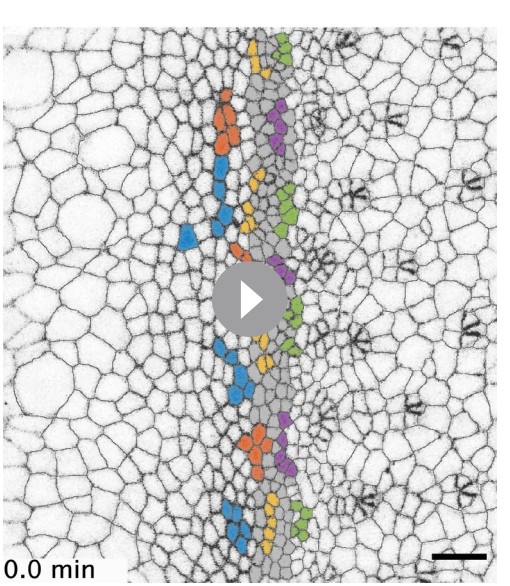

**Video 2.** A wildtype disc with the presumptive R cells labeled a specific color.
All R cells belonging to the same column are labeled with the same color. Cells shaded gray represent a strip of cells residing within the MF.
https://elifesciences.org/articles/72806/figures#video2

was previously observed and attributed to transcriptional activation of the *E-cadherin* gene by Atonal, and post-transcriptional activation of the gene by the Epidermal Growth Factor Receptor (EGFR) (*Brown et al., 2006*). In conclusion, the cultured eye discs exhibited features of development that eye discs in vivo have been inferred to have. This gives us confidence that our system is an accurate model of in vivo eye patterning.

## Periodic flows in the MF

Previous studies had inferred that cells in the eye disc do not significantly migrate or move with respect to one another (*Campos-Ortega and Hofbauer, 1977*; *Ready et al., 1976*). It was thought that as the MF transits, cells remain stationary within the apical plane and relative to each other, allowing the wavefront of gene expression pass through them. This assumption was fundamental to interpretations of later genetic experiments, leading to the reaction-diffusion model for ommatidia patterning embodied in present mathematical models (*Gavish et al., 2016*; *Lubensky et al., 2011*). We investigated the accuracy of this assumption by tracking individual cells within the MF and estimating the velocity of each cell over time. We observed an unexpected collective cell flow pattern in the MF directed toward the anterior (*Figure 2A*). Cells were observed to collectively flow together within clusters of 15–25 cells. Clusters flowed at a similar anterior velocity as other clusters, forming a periodic pattern of flow along the dorsal-ventral axis. The clusters with high anterior flow were moving at a similar velocity (1.2–1.5 µm/hr) to the MF, which moved at 1.44 µm/hr (*Figure 2—figure supplement 1*). Clusters with lower flow moved at 0.35–0.55 µm/hr. This periodic flow field was stable throughout the time course of imaging and was localized to the MF (*Figure 2B*). Strikingly, the period length of the flow field matched the average distance between neighboring ommatidia within the same column posterior to the MF (*Figure 2C*).

Although there was a stable spatially periodic flow field within the MF, the magnitude of the flow in each cluster was not constant over time — its phase varied. A cluster at a fixed position along the dorsal-ventral axis that experienced high flow would then experience low flow approximately 2 hr later, and vice versa (*Figure 2A and D*). Thus, all clusters oscillated in the magnitude of their flow over time with a regular period length of ~4 hr (*Figure 2E*), the same time taken to form two new columns of ommatidia. This observation suggested clusters in the MF that were added to even-numbered columns would flow out of phase with those clusters forming odd-numbered columns. We binned cells in the MF by virtue of their fates to form even or odd numbered columns, and we tracked their velocities in the MF over time (*Figure 2F*). Periodic fluctuations in velocity were synchronized between cells in the MF that were added to columns of the same parity. These fluctuations were out of phase to the velocity field that occurred in cells being added to columns of the opposite parity.

Given these observations and recent studies showing the importance of mechanics in follicle spacing in the vertebrate skin (*Glover et al., 2017*; *Shyer et al., 2017*), we began to develop a mechanical model for the formation of the lattice of ommatidia. First, we mapped the positions of presumptive R cells within the MF over time (*Figure 3A*). Each group of presumptive R cells moved to the anterior at a constant speed, but then abruptly stopped moving. At this inflection point, the group of five presumptive R cells were arranged in a linear configuration within the MF. Approximately two hours after they stopped moving, they had formed arcs and were clearly exiting the MF (*Figure 3A*). Presumptive R cells fated to form different columns moved at similar speeds to one another but stopped moving at phase-varied 2-hr intervals.

To relate these R cell dynamics to the periodic cell flows described earlier, we compared presumptive R8 cell velocities to other cells in the MF. Clusters of cells experiencing high anterior flow contained presumptive R8 cells that had comparable anterior velocities, suggesting that R cells were flowing coherently with the fast-moving clusters (*Figure 3C and D*, *Table 1* and *Video 3*). In contrast, clusters with lower flow contained presumptive R8 cells with divergent velocity behaviors (*Figure 3C and E*, *Table 1* and *Video 3*). Presumptive R8s on the anterior side of the MF were flowing coherently with the slow-moving clusters. However, presumptive R8s on the posterior side of the MF had stopped moving or were moving in a posterior direction. Thus, presumptive R8s residing in low-flow clusters had velocities that were both overlapping and distinct from other cells in these clusters (*Figure 3E*). When we considered all presumptive R8s in both high-flow and low-flow clusters within the MF, a clearly bimodal behavior in their anterior velocity was observed, and this was a stable property of the MF over the time course of imaging (*Figure 3F*, *Table 1*).

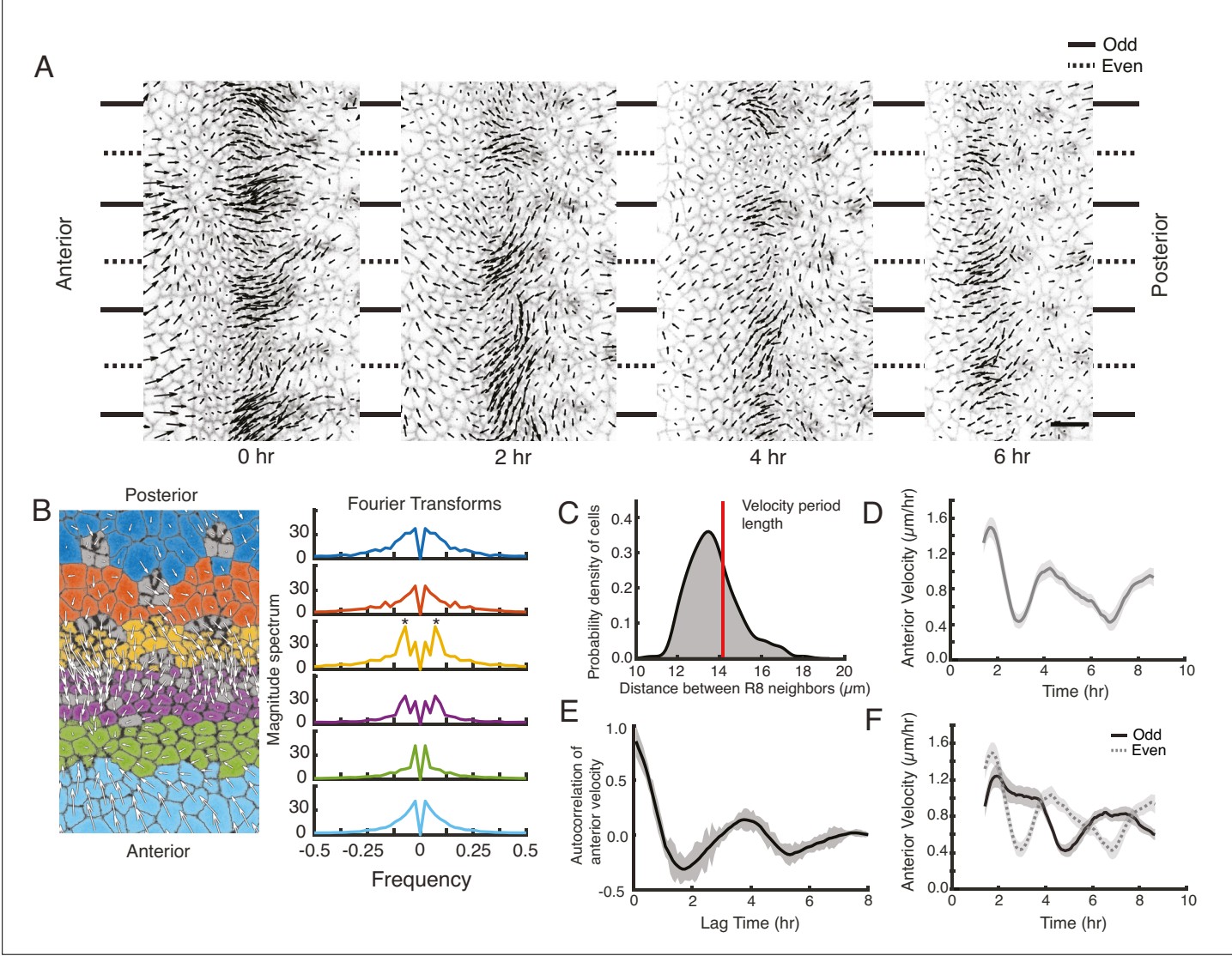

**Figure 2.** A periodic cell flow exists in the MF. (**A**) Velocities for all cells in the MF at the indicated times. Arrow length is proportional to the magnitude of velocity. Clusters of cells with higher anterior velocity alternate with clusters of lower anterior velocity, and their positions along the dorsal-ventral axis (top-bottom) correspond to where nascent ommatidia emerge to form even and odd numbered columns. Note that anterior velocities progressively diminish over the time course, which matches a corresponding diminishment in the rate of MF movement over the time course. The cause of progressive MF slowdown ex vivo is not known. Scale bar is 5 µm. (**B**) Cell velocities in the anterior direction were decomposed by Fourier transformation. Left, transformation was independently performed on cells binned by their anterior-posterior positions relative to the MF. Velocities for all cells in each bin were collected over five hours. Each color represents a single bin. Right, Fourier transforms of bins specific to the MF show anterior velocity oscillations that have a fixed spatial frequency (marked by asterisks). The peaks close to zero frequency seen in all transforms likely corresponds to the known equator-to-margin gradient in ommatidia maturation (***Wolff and Ready, 1993***). Ommatidia closest to the equator are the earliest to develop in a column, and marginal ommatidia are the last. (**C**) The distance between each pair of neighboring R8 cells within in a column was sampled for all columns of ommatidia and all time-frames. This is shown as a probability density function. The velocity oscillations detected by Fourier transformation have a fixed period length (14.17 µm, red line) that is highly comparable to the average distance between neighboring R8s. (**D**) Cells within the MF that reside at the same position along the dorsal-ventral axis exhibit temporal oscillations in anterior velocity over the time course of imaging. Shading represents 95% confidence intervals. (**E**) Autocorrelation of anterior velocity at time *t* for cells within the MF that reside at the same position along the dorsal-ventral axis. Velocity is maximally anticorrelated at *t* + 2 hr and is positively correlated with a maximum at *t* + 4 hr. These timescales are typical for formation of one and two new ommatidial columns, respectively. Shading represents 95% confidence intervals. (**F**) All cells within the MF that are fated to join odd-numbered columns of ommatidia synchronously oscillate their anterior velocity over time, and their velocity oscillates out of phase with cells fated to join even-numbered columns. Shading represents 95% confidence intervals.

The online version of this article includes the following figure supplement(s) for figure 2:

**Figure supplement 1.** Position of the MF as it changes over time.

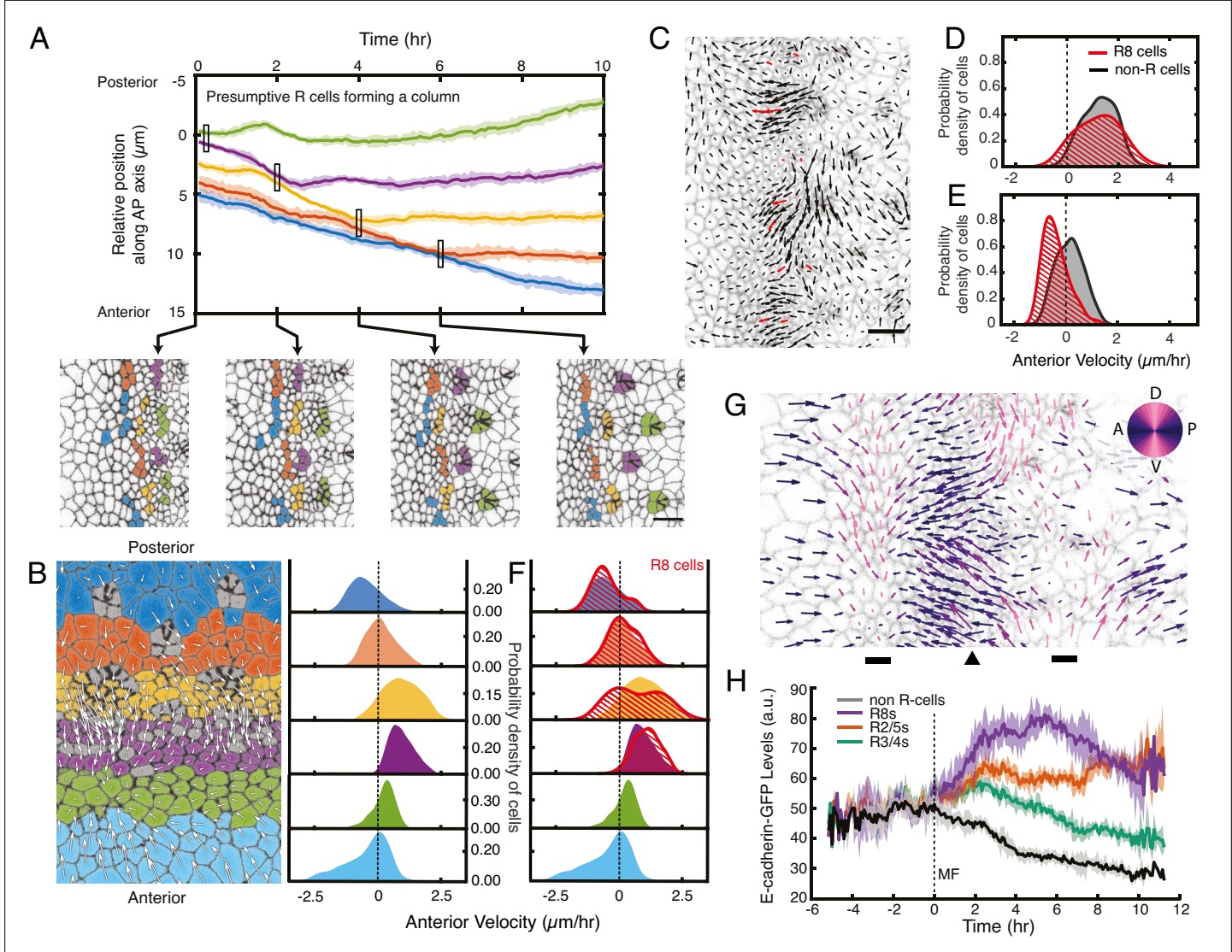

**Figure 3.** Complex cell flows within the eye disc. (**A**) Tracking over time the positions of all presumptive R cells binned by their ommatidia column. Each bin is color coded. The moving line averages follow relative position along the anterior-posterior axis of the eye disc. The position of cells in the first column that emerges from the MF is zero. Shading represents 95% confidence intervals. Below, segmented images at the indicated time points when some R cell bins inflect their velocity. (**B**) Distribution of cell velocities in the anterior direction. Left, analysis was independently performed on cells binned by their anterior-posterior positions relative to the MF. Velocities for all cells in each bin were collected over five hours. Each color represents a single bin. Right, probability density of cells in each bin with respect to their anterior velocity. (**C**) Time point showing velocities for the presumptive R8 cells in the MF (red) and all other cells (black). Note the presumptive R8s in each low-flow cluster can have velocities oriented in different directions from one other. (**D,E**) Probability density of presumptive R8s and non-R cells with respect to their anterior velocity. Cells are within MF clusters experiencing high flow (**D**) or low flow (**E**). All such cells were sampled over five hours. (**F**) Density distribution of anterior velocities of presumptive R8 cells over five hours. Cells were binned as in panel B. Note the bimodal distribution of R8 anterior velocities within the MF. (**G**) Time point showing velocities for all cells in the field of view color-coded by their relative direction. The MF (arrowhead) is flanked by two dorsal-ventral stripes of cells (bars) that have little to no velocity along the anterior-posterior axis (left-right). (**H**) Tracking of the average abundance of E-cadherin-GFP protein over time for all presumptive R cells and non-R cells. Shading represents 95% confidence intervals. Cells were synchronized in time by the point when they exited the MF (time 0). Note the uniform elevated levels of E-cadherin for all cells within the MF, and a bifurcation at time 0 when R cells increase E-cadherin and non-R cells decrease E-cadherin.

The online version of this article includes the following figure supplement(s) for figure 3:

**Figure supplement 1.** Triangular grid formation.

**Table 1.** Summary statistics of populations under comparison.

We performed non-parametric Mann-Whitney U tests since most distributions are not Gaussian. The large sample sizes going into all our measurements resulted in extremely low p-values for many. Therefore, we report the -log10(p-values) of the Mann-Whitney tests. The -log10 values report the statistical magnitude of the difference between one group and another; the larger that value, the larger the difference. For example, any -log10 value less than 1.3 means p > 0.05.

| Compared populations | -log10(Mann Whitney p-value) |
| --- | --- |
| *Figure 3D & E* - R8s vs. non-R cell flow in the MF | |
| 3D - R8 vs. non-R cells in high flow regions of the MF | 1.1467 |
| 3E - R8 vs. non-R cells in low flow regions of the MF | 24.8128 |
| *Figure 3F* - velocity distributions of R8s to non-R cells | |
| Dark blue (far posterior) | 0.5838 |
| Orange (PTZ) | 1.6628 |
| Yellow (posterior side MF) | 8.9340 |
| Purple (anterior side MF) | 15.5146 |
| *Figure 4D–D"* velocity distributions of strong *sca* vs. WT | |
| 4D top - R8s vs. non-R cells | 72.4072 |
| 4D bottom - R8s vs non-R cells | 23.9553 |
| 4D' top - Strong *sca* vs. WT | Infinity* |
| 4D' bottom - Strong *sca* vs. WT | Infinity* |
| 4D" top - Strong *sca* vs. WT | 68.3809 |
| 4D" bottom - Strong *sca* vs. WT | 22.5819 |
| *Figure 4H–H"* velocity distributions of weak *sca* vs. WT | |
| 4 H top - R8s vs. non-R cells | 12.7047 |
| 4 H bottom - R8s vs non-R cells | 20.9813 |
| 4 H' top - Weak *sca* vs. WT | 447.4232 |
| 4 H' bottom - Weak *sca* vs. WT | 24.8584 |
| 4 H" top - Weak *sca* vs. WT | 17.4521 |
| 4 H" bottom - Weak *sca* vs. WT | 0.3525 |
| *Figure 6B* - velocity distributions of R8s to non-R cells | |
| Far posterior | 0.5838 |
| Posterior Transition Zone (PTZ) | 1.6628 |
| Morphogenetic furrow (MF) | 8.9340 |

*p Value reported in the test lower than the computational limit of the program.

We next determined whether cells outside of the MF exhibited similar flow properties. The observed velocities across the entire field of view revealed complex mechanics (*Figure 3G*). Cells located immediately anterior to the MF primarily flowed in an anterior direction but without periodicity. This zone was positioned beside a narrow zone of cells that, on average, had little or no anterior-posterior movement, and anterior to that zone, cells appeared to be primarily moving in a posterior direction. Strikingly, cells posterior to the MF also flowed in a complex pattern (*Figure 3G*). Immediately posterior to the MF, cells flowed in an anterior direction. Further posterior was a narrow zone of cells that, on average, had little or no anterior-posterior movement, and posterior to that zone, all cells were moving in a posterior direction. These patterns of cell flow were a stable feature of the eye disc over the time course of imaging (*Figure 3B*).

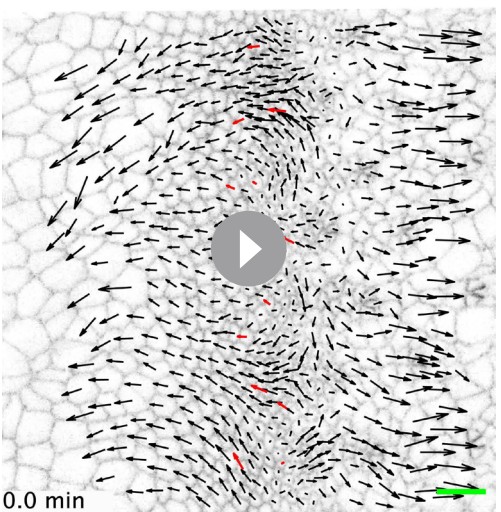

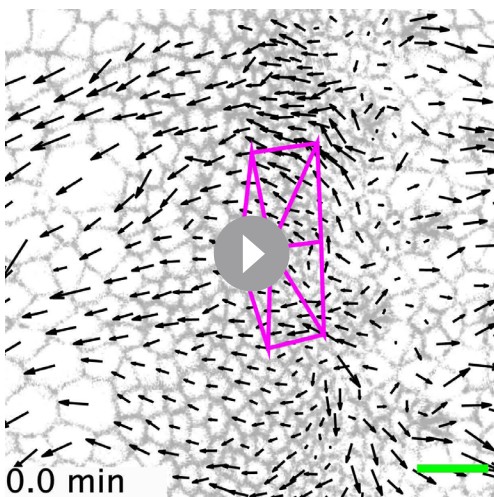

**Video 3.** A wildtype disc with all cells labeled with their respective velocity vectors, in which arrow length is proportional to the magnitude, and direction is marked by arrow orientation. Presumptive R8 cell velocities are labeled red.

https://elifesciences.org/articles/72806/figures#video3

**Video 4.** Emergence of the triangular lattice as a consequence of cell flow.
A group of seven presumptive R8 cells spanning five neighboring columns are connected to one another by purple lines. All cells are labeled with their respective velocity vectors. Note how the vertices of triangles move coherently with surrounding cell flows to create a regular triangular grid.

https://elifesciences.org/articles/72806/figures#video4

In sum, these observations led us to hypothesize that periodic cell flows contribute to the emergence of regularly spaced ommatidia. Reduced flow on the posterior side of the MF leads to groups of cells leaving the MF by virtue of having stopped moving at the same velocity as the furrow. A subset of cells in each group then form an ommatidium. Since the flow is periodic, it ensures that the groups of cells leaving the MF at any given point in time are regularly spaced apart. Moreover, the alternating phase of periodic flow in the MF over time ensures that groups of cells at the same dorsal-ventral position leave the MF every 4–5 hr or in every second column, thus generating a triangular grid pattern that is coherent with the already patterned portion of the lattice (*Figure 3—figure supplement 1*, *Video 4*).

## Template-driven regulation of cell fate

Emerging ommatidia are not only morphologically distinct, but they are undergoing gene expression programs that determine cell fates. If our hypothesis was correct, then a mechanical mechanism generates a lattice pattern of gene expression. The *atonal* gene is initially expressed uniformly in a dorsal-ventral stripe of cells in the MF that is resolved posteriorly into evenly spaced clusters of cells of decreasing number until *atonal* is expressed exclusively in presumptive R8 cells (*Frankfort and Mardon, 2002*). The Atonal proneural transcription factor then triggers a program of gene expression leading to R8 specification (*Pepple et al., 2008*). A hypothetical mechanical mechanism should generate regularly spaced clusters of Atonal-positive cells being left behind by the MF, while the other Atonal-positive cells continue moving with the MF. Since Atonal protein is localized to nuclei located far more basally than the adherens junction, we instead examined expression of a regulatory target of Atonal: the *E-cadherin* gene itself. Transcription of *E-cadherin* is strongly activated by Atonal in R cells as they exit the MF, while non-R cells downregulate *E-cadherin* transcription (*Brown et al., 2006*). Loss of *atonal* expression results in no upregulation of *E-cadherin* transcription in R cells (*Brown et al., 2006*).

Qualitatively, we could detect the noted regulation of E-cadherin-GFP expression in imaged cells (*Figure 1I* and *Figure 1—figure supplement 2A'*). Therefore, we measured E-cadherin-GFP fluorescence within every cell at each time point, and then binned cells according to the time relative to when they exited the MF (time 0). All cells within the MF expressed equivalent levels of E-cadherin (*Figure 3H*). When cells exited the MF, a bifurcation in E-cadherin expression dynamics appeared.

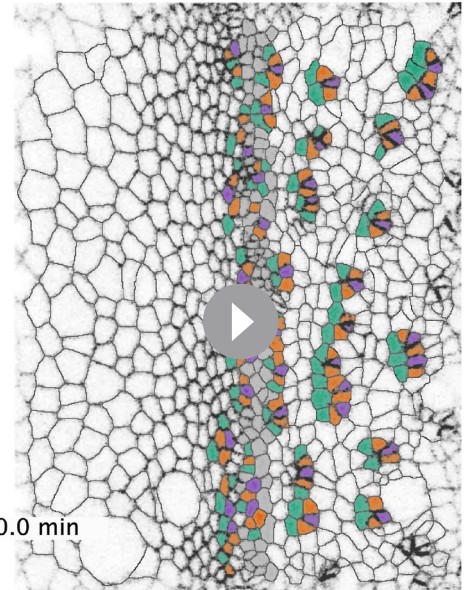

**Video 5.** A *scabrous* mutant disc with the presumptive R cells labeled with a specific color. R8 (purple), R2 and R5 (orange), R3 and R4 (cyan). Cells shaded gray represent a strip of cells residing within the MF. The disc has a more severe spacing phenotype.
https://elifesciences.org/articles/72806/figures#video5

Some cells progressively increased E-cadherin expression over time while other cells reduced their expression. The cells we had classified as being R cells based on apical shape and position were exclusively those that increased E-cadherin expression (*Figure 3H*). These dynamics are completely in accord with the known regulation of *E-cadherin* transcription by Atonal posterior to the MF (*Brown et al., 2006*). Thus, the mechanical process we were observing was generating a lattice pattern of proneural gene expression.

## Template-driven regulation of cell flow

An important prediction of our hypothesis is that any perturbation that disrupts the periodic cell flow would generate irregular spacing of ommatidia. Scabrous is a secreted glycoprotein that is required for regular spacing of ommatidia, and its loss results in irregular crowding of ommatidia (*Baker et al., 1990*; *Lee et al., 1996*). Given that Scabrous influences the regular spacing of ommatidia, we investigated whether it regulates the periodic flow of cells within the MF.

We imaged ex vivo eye discs from *scabrous*[BP2] null mutant larvae (*Video 5*). Cell division and delamination were comparable to wildtype, and the MF transited at a velocity comparable to wildtype (*Figure 4—figure supplement 1*). We found different mutant discs had spacing defects that ranged from mild to severe, as had been previously reported with fixed samples. Discs with strong defects showed an abnormally large number of ommatidia emerge from the MF to form each column (*Figure 4A*). Ommatidia clusters did not emerge with regular spacing, and frequently two neighboring clusters appeared to fuse together as they emerged. The *scabrous* mutant discs with strong spacing defects displayed a major impairment in flow within the MF. There was an almost complete absence of the periodicity in cell flow that had been observed in wildtype (*Figure 4B*). There were no discernible clusters of cells with similar anterior velocity (*Figure 4C*). Rather, cells all along the dorsal-ventral axis of the MF exhibited more uniform velocities (*Figure 4D*). When we compared the velocities of non-R cells in the MF between mutant and wildtype, mutant cells had comparable velocity to the average velocity observed within wildtype clusters having lower flow (*Figure 4D'*). A similar observation was made for the presumptive R8 cells; mutant cells displayed a loss of the wildtype bimodality in R8 velocity and uniformly resembled the wildtype R8s with low velocity (*Figure 4D"*). These differences were all significant (*Table 1*).

We also analyzed *scabrous*[BP2] mutant discs with mild spacing defects. These had an irregular triangular lattice structure due to occasionally misplaced ommatidia emerging from the MF (*Figure 4E*). These discs had a correspondingly milder impairment in the periodicity of cell flows in the MF (*Figure 4F and G*). Cells on the posterior side of the MF had a diminished flow compared to wildtype (*Figure 4H and H'*). Although presumptive R8 cells displayed bimodality in their velocity as with wildtype, the magnitude of velocity overall was reduced (*Figure 4H"*). These differences were significant (*Table 1*). Thus, *scabrous* not only perturbs both spacing and flow but it does so comparably for both phenomena.

We therefore posit that Scabrous regulates ommatidia spacing by inducing periodic clusters of cells in the MF to flow with it before slowing down to form a new column. The Scabrous protein is exclusively synthesized in R cells as they emerge from the MF, and protein expression is maintained in R8 cells within the first three columns (*Mlodzik et al., 1990*). Experiments with genetically

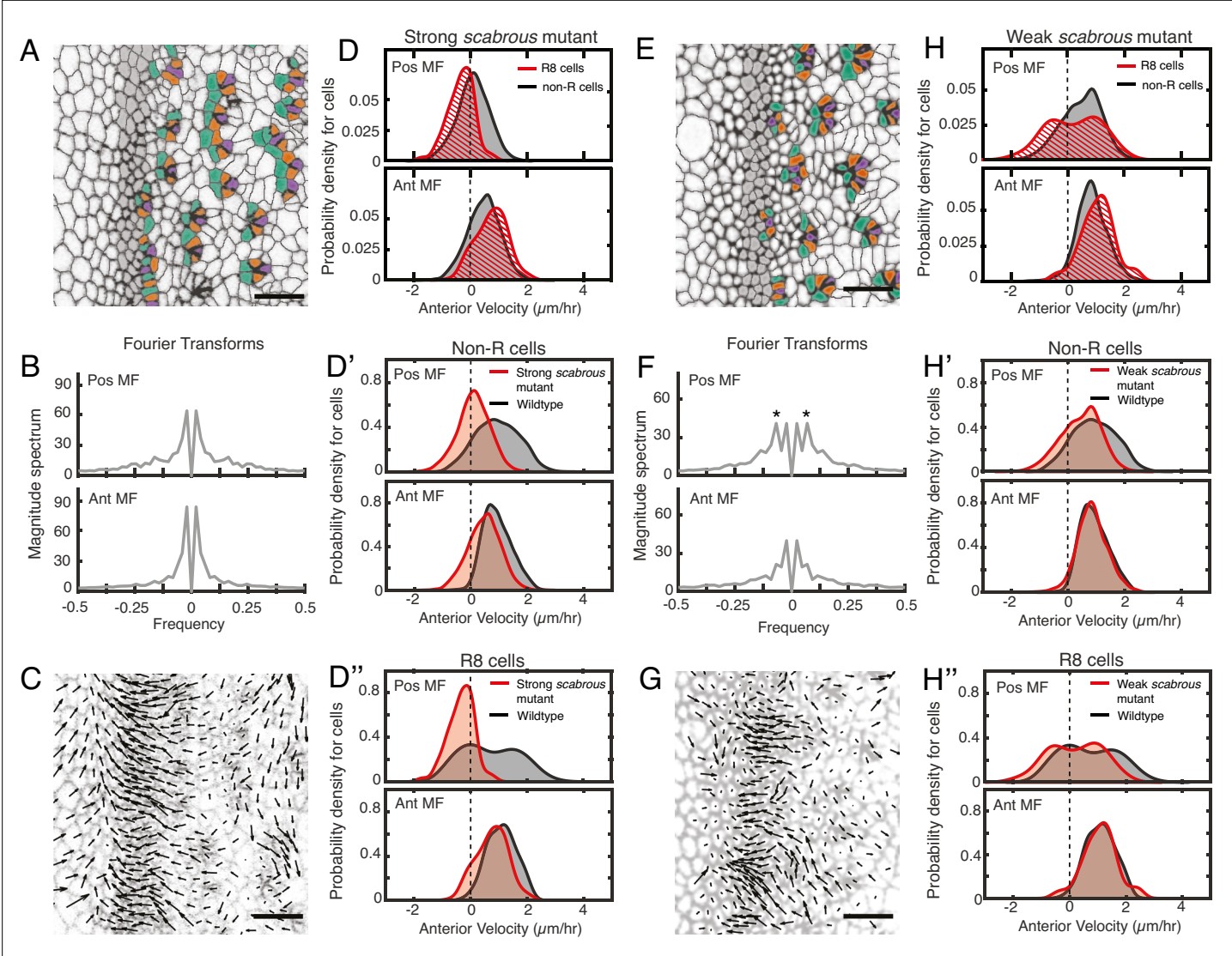

**Figure 4.** Scabrous is required for robust periodic cell flow in the MF. (**A-D″**) Analysis of *scabrous^BP2* eye disc with severe patterning defects. (**A**) Time point showing presumptive and fated R cells colored - R8 (purple), R2 and R5 (orange), R3 and R4 (cyan). (**B**) Cell velocities in the anterior direction were decomposed by Fourier transformation. Transformation was independently performed on cells binned by position relative to the MF, either on the posterior or anterior side of the MF. Velocities for all cells in each bin were collected over 5 hr. The peaks close to zero frequency correspond to the known equator-to-margin gradient in ommatidia maturation (*Wolff and Ready, 1993*). (**C**) A time point showing cell velocities. (**D-D″**) Probability density of presumptive R8 (red) and non-R (black) cells with respect to their anterior velocity. Cells were binned by their position along the posterior or anterior sides of the MF, and cells were sampled over five hours. (**D**) Comparison of presumptive R8s and non-R cells in the *scabrous* mutant. (**D′**) Comparison of non-R cells from *scabrous* mutant and wildtype eyes. (**D″**) Comparison of presumptive R8 cells from *scabrous* mutant and wildtype eyes. (**E-H″**) Analysis of *scabrous^BP2* eye disc with mild patterning defects. (**E**) Time point showing presumptive R cells colored as in A. (**F**) Cell velocities in the anterior direction were decomposed by Fourier transformation. Transformation was independently performed on cells binned by their position along the posterior and anterior sides of the MF. Velocities for all cells in each bin were collected over 5 hr. Velocity oscillations of fixed spatial frequency (marked by asterisks) are weaker than in wildtype (compare to *Figure 2B*). (**G**) A time point showing cell velocities. (**H-H″**) Probability density of presumptive R8 (red) and non-R (black) cells with respect to their anterior velocity. Cells were binned by their position along the posterior and anterior sides of the MF, and cells were sampled over 5 hr. (**H**) Comparison of presumptive R8s and non-R cells in the *scabrous* mutant. (**H′**) Comparison of non-R cells from *scabrous* mutant and wildtype eyes. (**H″**) Comparison of presumptive R8 cells from *scabrous* mutant and wildtype eyes.

The online version of this article includes the following figure supplement(s) for figure 4:

**Figure supplement 1.** Macroscopic features of *scabrous* mutant discs.

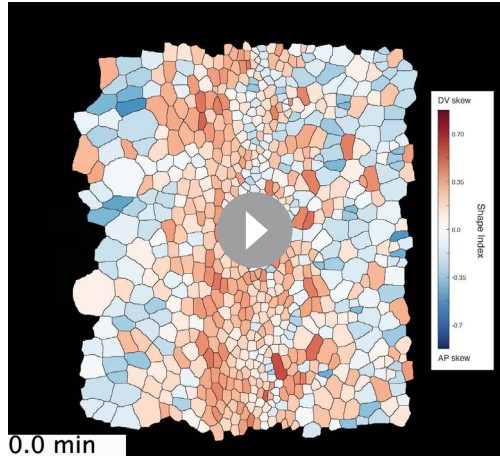

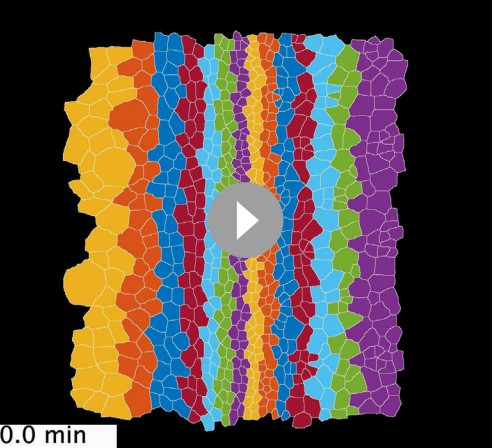

**Video 6.** A wildtype disc with all cells colored according to their Shape Index.
 Each cell at each time-frame was fitted with an ellipse, and the dorsal-ventral vs. anterior-posterior span of the ellipse was used to calculate the Shape Index of that cell. The Shape Index was defined as the (AP span - DV span)/(AP span+ DV span). The Shape Index for each cell is shaded such that cells with no anisotropy in their Shape Index are white and cells with anisotropy in either the DV or AP direction are shaded darker. If the Index indicated a larger span along the DV axis than AP axis, that cell is shaded orange. If the Index indicated a larger span along the DV axis than AP axis, that cell is shaded blue. Cells in the MF frequently have anisotropic Shape Indices with a larger span along DV axis than AP axis. Note that cells in the MF do not change their Shape Index or alignment to the body axes with any apparent periodicity.
https://elifesciences.org/articles/72806/figures#video6

**Video 7.** A wildtype disc with cells initially colored in arbitrary stripes of one to two cells thickness and that are aligned with the dorsal-ventral axis (vertical). Cells retain that color identity throughout the movie no matter where they move. Note that stripes of color within the anterior side of the MF retain their integrity, meaning that there are few T1 transitions pushing apart cells that are AP neighbors with one another. In contrast, frequent cell intercalations are observed posterior to the MF.
https://elifesciences.org/articles/72806/figures#video7

mosaic eyes indicate that Scabrous acts non cell-autonomously and must be expressed in R8 cells for normal patterning (*Baker et al., 1990*). Thus, newly formed ommatidia act as a chemical template to regulate strong periodic cell flows in an anterior direction within the MF in a manner dependent on Scabrous.

## Posterior cell dilation and MF flow

Given the complex pattern of non-homogeneous flow in the MF, we sought to understand their origins. Oriented rearrangements of cell shape and cell neighbors within the MF did not exhibit strong signatures of periodicity, and in fact cells within the MF showed few such rearrangements (*Videos 6 and 7*). This suggested that the origins of periodic flow are not intrinsic to the MF. However, we observed large-scale cellular contraction and dilation events occurring on the anterior and posterior sides of the MF, respectively (*Figure 5A*). Such events could in principle generate pressure gradients across the tissue, leading to cell flows. We quantified contraction and dilation as the rate of area change for each cell in the disc, and averaged these values for all cells at the same position along the anterior-posterior axis relative to the MF and across 5 hr of each movie. This one-dimensional perspective of the rate of change of apical area clearly showed that the rate of cell contraction was maximal in a narrow zone of cells located ~10 µm anterior to the MF, while the rate of dilation was maximal in cells located ~10 µm posterior (*Figure 5B*). Cells in these two zones corresponded to cells that we had previously observed were on average not flowing in either an anterior or posterior direction (*Figure 5C*). On either side of the anterior zone of contractility, cells flowed toward the zone, while on either side of the posterior zone of dilation, cells flowed away from the zone. These trends are consistent with pressure gradients originating along bands of cells anterior and posterior to the MF, leading to the complex cell flows observed in the eye disc.

To confirm that dilation in the posterior zone is coupled to flow, we examined cell dilation in the posterior region of *scabrous* mutants. We previously found that discs from *scabrous* mutants had diminished anterior flow in the MF (*Figure 4D' and D''*). In *scabrous* mutants, cells posterior to the

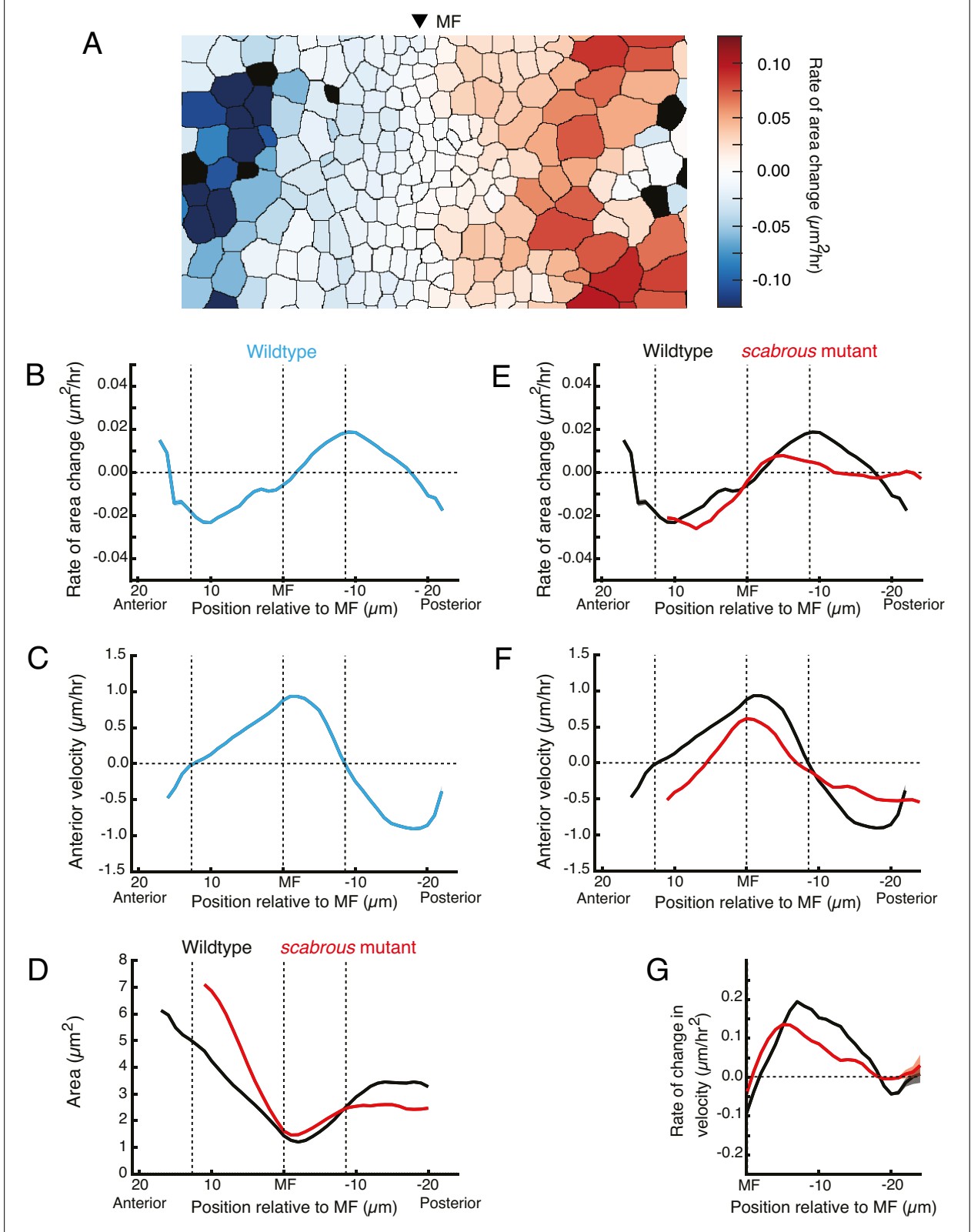

**Figure 5.** Scabrous is required for normal cell dilation and cell flow. (**A**) The rate of area change for all cells in the field of view at a randomly chosen time point. A stripe of cells anterior to the MF shows rapid area contraction, while a stripe of cells posterior to the MF shows rapid area expansion. (**B–G**) Measurements of various cell features as a function of cell position along the anterior-posterior axis relative to the MF. The MF position is fixed at zero. Measured features were averaged across all cells along the dorsal-ventral axis and over 5 hr of imaging, giving a one-dimensional representation.

*Figure 5 continued on next page*

Figure 5 continued

Analysis of all discs was aggregated such that each moving line average shown is representative for multiple samples. Shading represents 95% confidence intervals. Note these features are highly stable throughout the time course of imaging and between samples. For B through F, dotted vertical lines denote the stable positions of inflection points in wildtype cell area change and direction of cell movement. (**B**) Rate of area change in wildtype cells. (**C**) Anterior velocity in wildtype cells. (**D**) Cell area in wildtype and *scabrous* mutant cells. (**E**) Rate of area change in wildtype and *scabrous* mutant cells. (**F**) Anterior velocity in wildtype cells and *scabrous* mutant cells. (**G**) Rate of velocity change in wildtype and *scabrous* mutant cells posterior to the MF.

MF dilated to an average size that was 25–33% lower than in wildtype (***Figure 5D***). This was due to a marked reduction in the rate of dilation for *scabrous* mutant cells posterior to the MF (***Figure 5E***). The maximal rate of dilation was reduced by 75% and occurred closer to the MF than wildtype. In contrast, cell contraction dynamics anterior to the MF were identical to wildtype (***Figure 5E***).

We hypothesized that if cell dilation in the posterior zone is generating pressure gradients, then in *scabrous* mutant discs, we would observe reduced anterior flow on the anterior side of the zone and reduced posterior flow on the posterior side. This expectation was confirmed in the mutant discs (***Figure 5F***). As well, the rate of velocity change of cells away from the posterior zone was also diminished, consistent with an abatement in the forces generating flow (***Figure 5G***). Finally, the inflection point of cell flow in *scabrous* mutant discs was shifted closer to the MF, consistent with the anterior shift in the zone of maximal dilation rate (***Figure 5E and F***). These correlations between different features in the mutant discs lend further support to the hypothesis that the posterior dilation zone generates a pressure gradient leading to cell flows.

## Generation of periodic flow in the MF

Although posterior cell dilation would lead to anterior flow in the MF due to a pressure gradient, it alone cannot explain how the MF flow is periodic. One possibility is that the pressure gradient itself periodically fluctuates in magnitude along the dorsal-ventral axis (***Figure 6A***). To test for the presence of a periodic pressure gradient, we looked at cell flows on the posterior side of the dilation zone. If there was an intrinsically periodic pressure gradient, we would observe a bimodal distribution of velocities on the posterior side as well as on the anterior side. This was not observed; only the anterior side exhibited strong bimodality in velocity (***Figure 6B***, ***Table 1***).

Pressure gradients in viscous fluids generate flow fields. The emergent flow profile generated, however, must accommodate boundaries, suggesting that a periodic flow profile can be generated by a uniform pressure gradient and a periodicity in the location of boundaries. We considered a mechanism whereby ommatidia in the act of formation become immobile within the flow field anterior to the dilation zone, forming a periodic array of rigid boundaries. (***Figure 6C***). These rigid boundaries could deflect and focus the flow around them generating periodicity in the flow. This mechanism would require ommatidia to be motionless relative to non-ommatidial cells in the zone where a periodic flow profile is observed. We averaged velocities for all presumptive and determined R cells over half of the imaging session, and plotted these as a function of their anterior-posterior position relative to the MF (***Figure 6D***). The presumptive R8 cells rapidly decreased their velocity as they emerged from the MF and they stopped moving when they were positioned several µm anterior to the position where all other cells also stopped moving. The other R cells in each rosette also slowed down far earlier than non-ommatidial cells. As ommatidia matured to become members of the third column, they reached the zone where all cells stopped moving (also the zone of maximum dilation). There, all of the R cells and non-ommatidial cells began to flow in the posterior direction at the same velocity (***Figure 6D***). Thus, ommatidia in the first three columns are immobile relative to the non-R cell flow field.

This regulation of R cell mechanics had not been observed before. The mechanism behind R8 immobility might involve a stable attachment to the basement membrane of the epithelium. We tested for the necessity of Scabrous to elicit R8 immobility. We measured velocity profiles of maturing R cells in *scabrous* mutant discs and found them to be highly similar to wildtype (***Figure 6E***). Therefore, Scabrous expression in R8 cells is not responsible for the transient immobility of ommatidia.

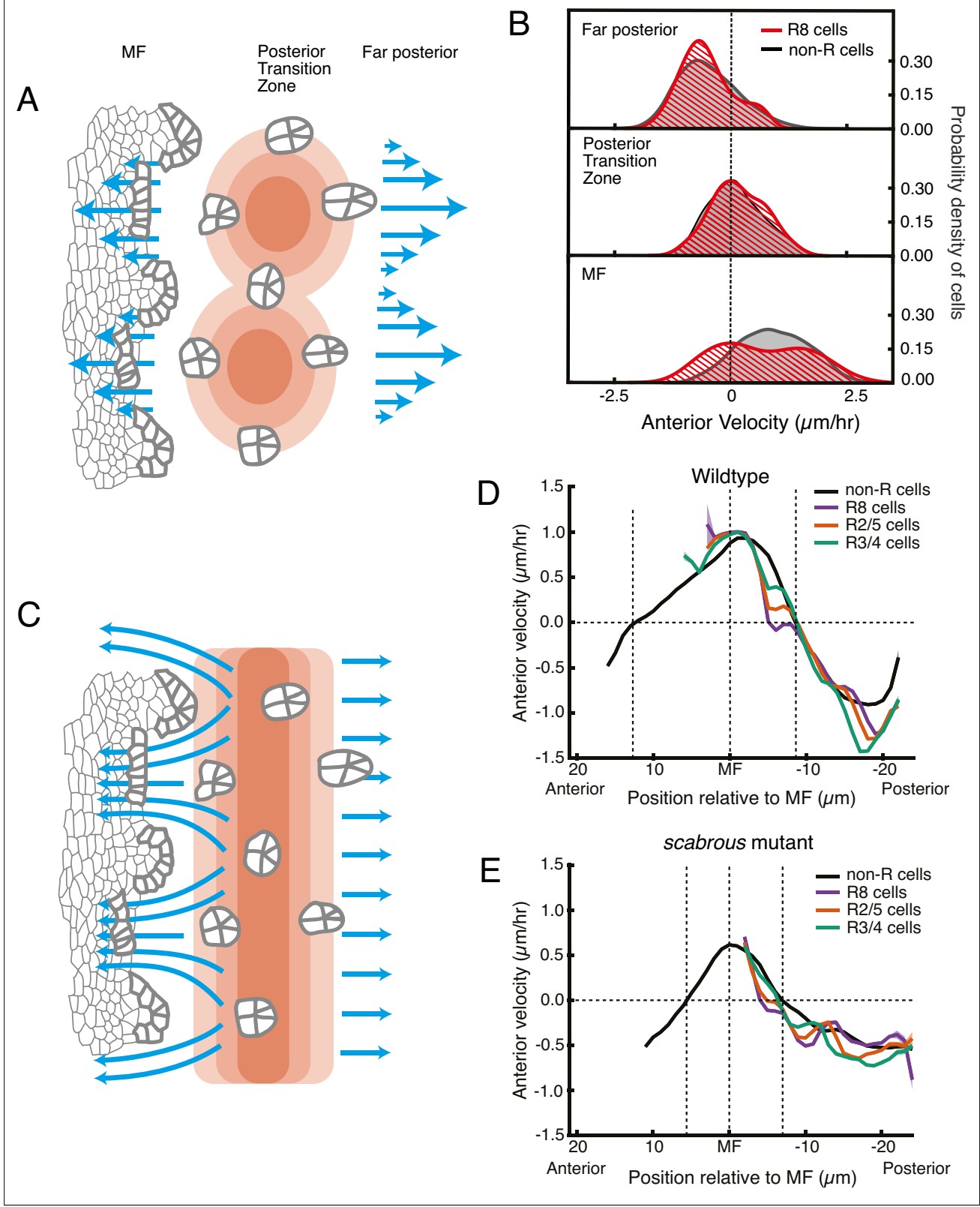

**Figure 6.** Relationship between cell flow and cell dilation/contraction. (**A**) Hypothetical mechanism for how an inferred pressure gradient generates periodic flows in the MF. There are periodic domains of cell dilation (red) in the source of cell flows, causing periodic flow (blue arrows) away from the source in both posterior and anterior directions. (**B**) Probability density of presumptive R8 (red) and non-R (black) cells with respect to their anterior velocity. Cells are binned according to their positions relative to the source of cell flow. Velocities for all cells in each bin were collected over 5 hr. (**C**)

*Figure 6 continued on next page*

*Figure 6 continued*

Alternative mechanism for how an inferred pressure gradient generates periodic flows in the MF. The zone of cell dilation is uniform, causing uniform flow in both posterior and anterior directions. However, if ommatidia cells are immobilized, then they cause cells to flow around them, leading to a periodic flow field in the MF. (**D,E**) Measurements of anterior velocity of cells as a function of cell position along the anterior-posterior axis relative to the MF. The MF position is fixed at zero. Velocity was averaged across all cells along the dorsal-ventral axis and over five hours of imaging, giving a one-dimensional representation. Shading represents 95% confidence intervals. For both wildtype (**D**) and the severe *scabrous* mutant (**E**) discs, R cells abruptly decrease in velocity immediately posterior to the MF. R8 cells remain immobile until all other cells reach the velocity inflection point at the Posterior Transition Zone, marked with a vertical dotted line.

## Discussion

In summary, we have described a complex set of phenomena never before observed in development of the *Drosophila* eye (*Figure 7A*). There is a source and sink of cell flow that straddles the MF. The source and sink are narrow stripes of cells oriented parallel to the MF and located at fixed distances from the MF. They generate complex cell flows across the eye disc, including periodic flow in the MF. As ommatidia emerge from the MF, they are initially immobile and do not flow with their non-R cell neighbors. When the immobile ommatidia reach the flow source posterior to the MF, they abruptly become mobile and begin to flow with surrounding cells.

These observations strongly argue against the long-standing model for ommatidia patterning, which invoked local cell-cell interactions directly regulating the R cell gene expression program to create a lattice prepattern in the wake of the MF (reviewed in *Frankfort and Mardon, 2002*; *Roignant and Treisman, 2009*). That model relied upon all cells remaining immobile with respect to one another as the MF passes, so that long-range lateral inhibitors emanating from newly formed ommatidia would establish stable zones of non-neural fate (*Figure 7B*). Instead, we have found that cells within the MF significantly move with respect to one another, a condition that precludes a standard reaction-diffusion mechanism from creating a regular prepattern of gene expression.

Our observations instead support a mechanochemical mechanism for ommatidia patterning. Groups of cells within the MF flow at the same velocity as the MF, and they alternate with groups of cells flowing at a much lower velocity. These latter groups are left behind on the posterior side of the MF and form a new column of ommatidia. Flow in the MF is generated by the source and sink that straddle the MF. The posterior source of cell flow precisely corresponds to a zone of cell dilation. We suggest the dilation zone creates a pressure gradient, which causes cells in the MF to flow away from the source (*Figure 7C*). The anterior sink of cell flow precisely corresponds to a zone of cell contraction, suggesting that it contributes to the pressure gradient and augments cell flow in the MF towards the sink (*Figure 7A*). Flow in the MF is non-uniform owing to the lattice grid of immobile ommatidia cells deflecting and focusing the flow of non-ommatidia cells (*Figure 7A and C*). The resulting fluid flow is significantly focused in the MF to create a periodic flow field. In this sense, the templating mechanism is mechanochemical in origin; newly formed columns of differentiating ommatidia serve as a template to spatially focus cell flow that will move cells into position to form the next column of ommatidia with the opposing parity.

New ommatidia are undergoing a gene expression program to determine cell fates (*Chen and Desplan, 2020*). How does a mechanical mechanism generate a regular pattern of gene expression? Feather follicles in the avian skin are shaped by self-organizing mechanics, which directly induces the follicle gene expression program via cellular mechanosensation (*Shyer et al., 2017*). Although it is possible that eye disc cells employ mechanosensation as well, previous findings suggest a different mechanism might be at work (*Chen and Desplan, 2020*; *Roignant and Treisman, 2009*). BMP and Hedgehog signals induce cells entering the MF to express the proneural gene *atonal*. A dorsal-ventral stripe of cells in the MF express this proneural transcription factor (*Figure 7C*). The mechanism we describe would lead to regularly spaced clusters of Atonal-positive cells being left behind by the MF, while the other Atonal-positive cells would continue moving with the MF. Thus, a striped *atonal* prepattern would be resolved to a lattice pattern of *atonal* expression by tissue mechanics.

The posterior source of cell flow is a stripe of cells that undergo a set of coordinated yet diverse morphological changes (*Figure 7A and C*). Hence, we are naming this stripe the posterior transition zone (PTZ). Non R cells in the PTZ are experiencing maximal dilation. Ommatidia in the PTZ abruptly change their mobility relative to other cells, going from immobility to ambient mobility.

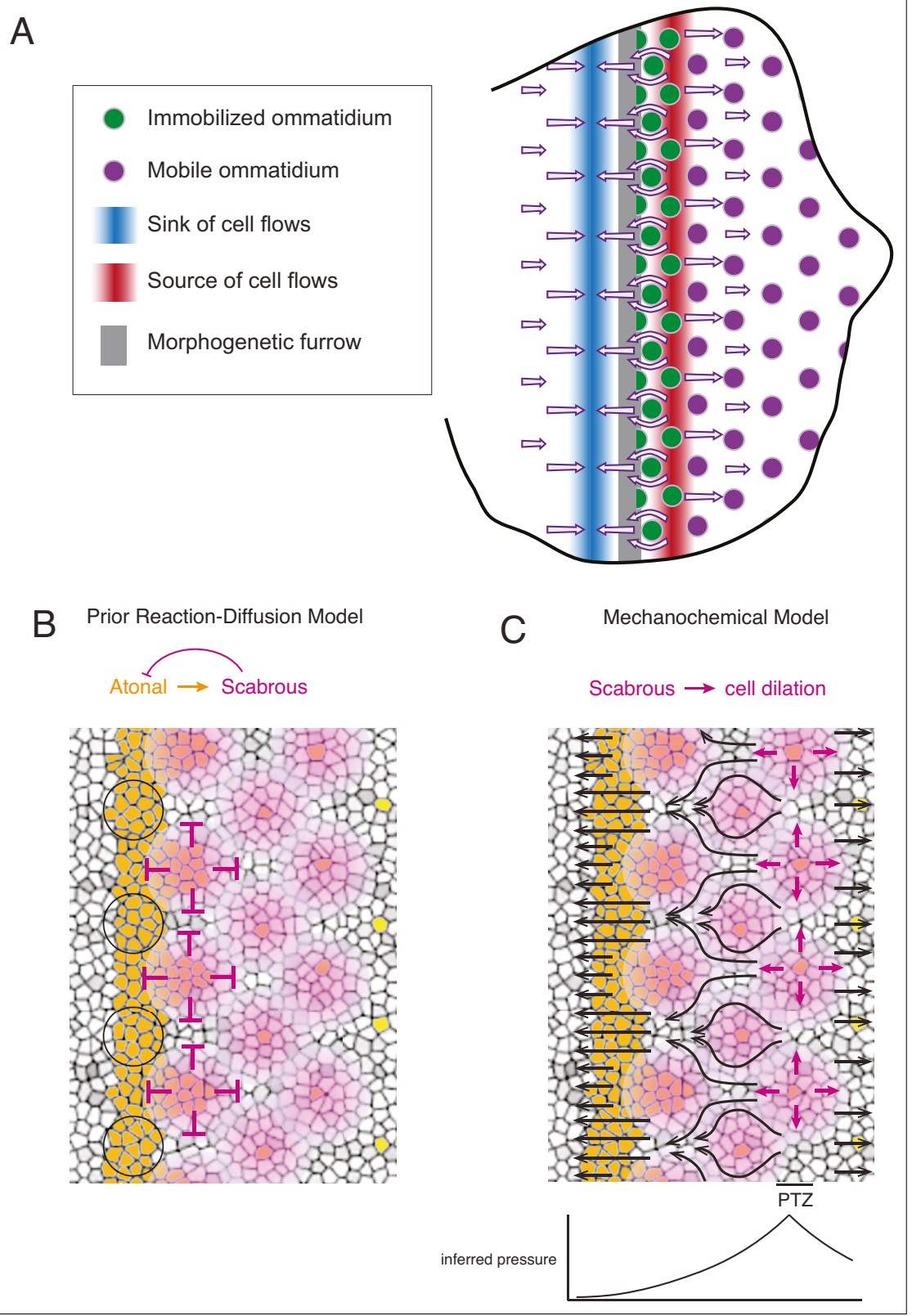

**Figure 7.** Self-organized triangular lattice formation. (**A**) Summary of newly observed phenomena described in this study. Arrows show the cell flows across the eye disc. (**B**) Prior model for self-organized lattice formation based on a standard reaction-diffusion process. It was thought the stripe of Atonal-positive cells resolves into a triangular grid of R8 cells due to inhibitory fields of Scabrous diffusion (purple). Alternating clusters of Atonal-positive cells in the MF (black circles) lie outside the inhibitory fields and therefore maintain Atonal expression and become the next column of

*Figure 7 continued on next page*

*Figure 7 continued*

ommatidia. (**C**) Mechanochemical model for self-organized lattice formation. R8 cells secrete Scabrous in columns one to three, triggering nearby cells to dilate. At the posterior transition zone (PTZ), the rate of dilation is maximal, creating a pressure gradient. Since ommatidia cells in columns one to three are immobilized, anterior cell flow down the pressure gradient is deflected and focused to create flow in the MF that is periodic. This leads to alternating clusters of cells flowing with the MF before slowing down and exiting the MF.

Posterior to the PTZ, ommatidia begin their rotation, and presumably the release from immobility enables rotation to occur. An important point to emphasize is that the PTZ remains a fixed distance from the MF as it migrates across the disc. This would be crucial to maintain a constant pressure gradient toward the MF such that periodic flows would generate an invariant triangular grid of ommatidia across the eye field. The fixed distance also ensures that the pattern of immobile ommatidia in columns one and two focuses cell flow into alternating periodic clusters within the MF as it moves.

Scabrous is a glycoprotein secreted by R8 cells only in the first three columns of ommatidia behind the MF. Prior studies suggested that Scabrous might be either a negative or positive signal for patterning gene expression (*Frankfort and Mardon, 2002*; *Gavish et al., 2016*; *Lubensky et al., 2011*). We found that loss of Scabrous leads to diminished cell dilation in the PTZ, diminished cell flow, and broken emergence of the ommatidia lattice at the MF. One scenario is that Scabrous directly acts in parallel on each of these processes, which are independent of one another (correlated without causal connection). It is difficult to imagine how a single protein could independently elicit such diverse cell behaviors as differentiation and cell size. A more likely scenario is that Scabrous directly acts on one of these processes and indirectly on the others – and the one directly regulated process regulates the others (correlated with causal connection). If so, there are two possible pathways. One, the broken lattice is upstream of the defects in dilation and flow, meaning the defects in spacing directly caused by the mutant have consequences on dilation and flow more posterior. The second possibility is the dilation and flow are upstream of the broken lattice, meaning that the mutant directly impacts dilation/flow that has consequences on lattice formation.

We favor the second scenario because the first scenario still leaves unresolved how Scabrous could pattern the lattice solely by regulating gene expression. A fundamental and essential pre-requisite for Scabrous to work that way is for cells in the epithelium not to move relative to one another. We have shown that such an assumption is false – cells move a lot. Thus, for Scabrous to work as a differentiation signal, it would need to signal nearby cells to regulate their gene expression, and then these cells would flow with some periodicity into position and differentiate. Scabrous would not work by classic reaction-diffusion but by some hybrid mechanism. Formally, such a mechanism is possible though complicated. We favor the simplest model in which Scabrous directly regulates cell dilation and consequently cell flow and patterning. Note that this model fits with all previously published genetic data on Scabrous' role in eye development.

It was suggested that Scabrous acts as a ligand for the Notch receptor (*Powell et al., 2001*), but this suggestion was not substantiated in the eye (*Lee et al., 1996*; *Roignant and Treisman, 2009*). If Scabrous is a receptor ligand, the identity of the signal transduction pathway remains unknown. Perhaps such a pathway triggers cell dilation. Alternatively, Scabrous might stimulate dilation more directly. It is also possible that Scabrous promotes cell flow by weakening attachments between flowing cells with one another or with the extracellular matrix, but this would necessitate two distinct processes Scabrous would control: cell dilation and attachment. Interestingly, *scabrous* mutants have a small but significant phenotype in which their ommatidia over-rotate (*Chou and Chien, 2002*). Although the mechanism behind this was not elucidated, it is possible the Scabrous-induced posterior cell flow somehow brakes rotation. Understanding the molecular mechanism of Scabrous will help elucidate these questions.

We found that null *scabrous* mutants had variable and partial effects on dilation and flow that correlated with their effects on ommatidia spacing. Variability in spacing phenotypes for the null *scabrous* mutant had been previously noted (*Baker and Zitron, 1995*). There is another signal important for spacing that is induced by EGFR (*Chen and Chien, 1999*; *Spencer et al., 1998*). This signal must act in parallel to Scabrous for ordering the triangular grid pattern, since a *scabrous EGFR* double mutant has more severe spacing defects than either single mutant (*Baonza et al.,*

2001). Therefore, it is possible that EGFR signaling also contributes to the pressure and flow fields that we have discovered.

# Materials and methods

## Key resources table

| Reagent type (species) or resource | Designation | Source or reference | Identifiers | Additional information |
|---|---|---|---|---|
| Gene (*Drosophila melanogaster*) | *DE-Cadherin::GFP* | *Huang et al., 2009* | Flybase: FBal0247908 | Knock-in in-frame fusion of GFP into endogenous *shotgun* (*shg*) gene. Gift from Suzanne Eaton |
| Gene (*Drosophila melanogaster*) | *scabrous*^BP2^ | Bloomington *Drosophila* Stock Center | FlyBase ID FBal0032653 BDSC ID 7320 | ~ 2 kb deletion of 5' region of *sca*, including first exon and start site. Protein null allele. |
| Other | Grace's Insect Medium | Sigma Aldrich | G9771 | |
| Chemical compound, drug | PenStrep Stock Solution | Gibco | 15140–122 | |
| Chemical compound, drug | BIS-TRIS | Sigma Aldrich | B4429 | |
| Chemical compound, drug | Fetal Bovine Serum (FBS) | Thermo Fisher Scientific | 10270098 | |
| Chemical compound, drug | Insulin solution, human | Sigma Aldrich | 19,278 | |
| Other | Dumont forceps (0208–5-PO) | Fine Science Tools | 11252–00 | |
| Other | 26G × 5/8 Syringe | BD | 305,115 | |
| Other | 35 mm dish, No. 1.5 coverslip | MatTek | P35G-1.5–14 C | |
| Chemical compound, drug | 0.01% (w/v) Poly-L-Lysine | Sigma Aldrich | P4707 | |
| Other | Tesa double sided sticky tape | Tesa | 5,338 | |
| Other | 0.25 in (6 mm) hole puncher | Staples | 10,573 CC | |
| Other | Whatman polycarbonate membrane | Sigma Aldrich | WHA70602513 | |
| Chemical compound, drug | SeaKem Gold Agarose | Lonza Rockland | 50,150 | |
| Software, algorithm | ImSAnE MATLAB software | *Heemskerk and Streichan, 2015* | | https://github.com/idse/imsane |
| Software, algorithm | Linear stack alignment with SIFT ImageJ plugin | *Lowe, 2004* | | https://imagej.net/plugins/linear-stack-alignment-with-sift |
| Software, algorithm | MATLAB implementation of Hungarian algorithm | *Cao, 2021* | | https://www.mathworks.com/matlabcentral/fileexchange/20328-munkres-assignment-algorithm |
| Software, algorithm | Convolutional neural network used for pixel classification | This paper | | https://github.com/K-D-Gallagher/CNN-pixel-classification |

*Continued on next page*

*Continued*

| Reagent type (species) or resource | Designation | Source or reference | Identifiers | Additional information |
|---|---|---|---|---|
| Software, algorithm | Trained CNN model for pixel classification of epithelial fluorescence confocal data | This paper | | https://drive.google.com/drive/folders/1I-nRpn1esRzs5t4ztgbNvkBQuTN2vT7L?usp=sharing |
| Software, algorithm | MATLAB pipeline for segmenting cells from pixel classified images and tracking them | This paper | | https://github.com/K-D-Gallagher/eye-patterning |
| Software, algorithm | MATLAB GUI for manually correcting segmentation errors | This paper | | https://github.com/K-D-Gallagher/eye-patterning |
| Software, algorithm | MATLAB datasets for Wildtype 1, Wildtype 2, Strong *scabrous* mutant, and Weak *scabrous* mutant | This paper | | https://drive.google.com/drive/folders/1I-nRpn1esRzs5t4ztgbNvkBQuTN2vT7L?usp=sharing |

## Experimental model and subject details

For all experiments, *Drosophila melanogaster* was raised using standard lab conditions and food. Stocks were either obtained from the Bloomington Stock Center or from listed labs. A list of all stocks used in this study is in the Key Resources Table.

## Genetics

The E-cadherin-GFP allele used for all studies was *DE-Cad::GFP* (*Huang et al., 2009*), a gift from Suzanne Eaton. This allele has GFP inserted into the endogenous E-cadherin gene *shotgun*. The allele

**Table 2.** Summary of published conditions to culture *Drosophila* imaginal discs ex vivo. Adapted from *Tsao et al., 2016*.

| Reference* | Medium | Serum (v/v) | Antibiotic | Fly extract (v/v) | Insulin | Ecdysone | Sample type | Duration |
|---|---|---|---|---|---|---|---|---|
| *Robb, 1969* | R-14 | --- | 1X | --- | --- | --- | wing disc | --- |
| *Davis and Shearn, 1977* | X (XCS) | --- | --- | --- | 0.4 mU/ml | 1 ng/ml | discs | --- |
| *Wyss, 1982* | ZW | ??? FBS | --- | 22.5% | 10 mg/ml | 10 ng/ml | disc cells | --- |
| *Currie et al., 1988* | Shields and Sang M3 | 2% FBS | --- | 5% | 125 mU/ml | 1 ng/ml | disc cells | --- |
| *Schubiger and Truman, 2000* | Shields and Sang M3 D22 | 7.5% FCS | --- | --- | --- | 1 mg/ml | wing disc | 24 hr |
| *Gibson et al., 2006* | Shields and Sang M3 | 10% FBS | 1X | --- | 0.01 mU/ml | --- | | 1.5–2 hr |
| *Cafferty et al., 2009* | Schneider's | 1% FBS | 10 X | --- | 200 mg/ml | --- | | 4 hr |
| *Landsberg et al., 2009* | Shields and Sang M3 | 2% FCS | 1X | 2.5% | 0.125 iu/ml | --- | wing disc cells | |
| *Aldaz et al., 2010* | Shields and Sang M3 | 2% FCS | 5X | --- | --- | 100–500 ng/ml | disc eversion | 10 hr |
| *Mao et al., 2011* | Shields and Sang M3 | 2% FBS | 1X | --- | 0.01 mU/ml | --- | wing disc | 5 hr |
| *Ohsawa et al., 2012* | Schneider's | 10% FBS | --- | --- | --- | --- | | 3 hr |
| *Zartman et al., 2013* | Schneider's | variable | 4X | 5% | 6.2 µg/ml | --- | wing disc | 5 hr |
| *Legoff et al., 2013* | Shields and Sang M3 | 2% FCS | --- | 2.5% | 125 mU/ml | --- | wing disc | 8 hr |
| *Handke et al., 2014* | Shields and Sang M3 | 2% FBS | 10 X | 5% | 5 µg/ml | 1 ng/ml | wing disc | 7 hr |
| *Tsao et al., 2016* | Schneider's | 2% FBS | 4X | --- | 1250 µg/ml | --- | multiple discs | 18 hr |
| *Dye et al., 2017* | Grace's Insect Medium | 5% FBS | 1X | --- | --- | 1 ng/ml | wing discs | 12+ hr |
| This study | Grace's Insect Medium | 2% FBS | 1X | --- | 625 µg/ml | --- | eye disc | 12–16 hr |

was created by homologous recombination within a founder line containing a deletion of the *shotgun*. The recombination restored the intact *shotgun* sequence with a C-terminal GFP tag. While this allele does contain an attR sequence in a non-conserved intronic region, it was ascertained that its expression levels are not significantly different than wildtype E-cadherin levels. The *scabrous* (*sca*) mutant allele used for all studies was *sca*[BP2] (*Mlodzik et al., 1990*). This allele has a 2 kb deletion of the 5′ end of the transcription unit, removing the transcription start site, the 5′ UTR, and the beginning of the open-reading frame (*Baker et al., 1990*). It is a genetic amorph and is null for protein expression. The *sca*[BP2] allele was recombined with the E-cadherin::GFP allele, as both genes are located on the right arm of the second chromosome.

## Media composition for disc culture

Most published studies of ex vivo imaginal disc culture have used either Schneider's media or Shields and Sang's M3 media supplemented with standard concentrations of fetal bovine serum (FBS), insulin, fly extract, and 20-hydroxyecdysone (ecdysone) (*Table 2*). Imaginal discs incubated under these culture conditions undergo rapid mitotic arrest and compromised gene expression (*Handke et al., 2014*). However, three recent advances in ex vivo culturing conditions have helped accelerate the discovery of optimal conditions for eye imaginal disc culture. (1) Basal media choice sensitizes the discs to different additives; that is Schneider's media sensitizes discs to insulin, whereas Shields and Sang's M3 sensitizes discs to fly extract (*Zartman et al., 2013*). (2) Extremely high concentrations of human insulin (1,250 µg/ml) support proliferation/development in the eye disc (*Tsao et al., 2016*). (3) Wing discs are more mechanically stable in Grace's insect medium (*Dye et al., 2017*). Relying upon these above-mentioned developments, we tested a variety of growth media types and concentrations/combinations of additives (FBS, insulin, and ecdysone). We found eye imaginal disc explants to be most mechanically stable in Grace's insect medium. Additionally, a more moderate concentration of insulin (625 µg/ml) than the *Tsao et al., 2016* protocol supported proliferation and eye disc development for 12–16 hr. Finally, ecdysone did not have any effect on eye disc development. The composition of our culture media is 1 x Grace's insect medium (Sigma, G9771) prepared without sodium bicarbonate, 1 x PenStrep (Gibco 15140–122), 5 mM BIS-TRIS (Sigma B4429), 2% (v/v) FBS (ThermoFisher 10270098), and 625 µg/mL human insulin (Sigma I9278) (*Table 2*).

Culture media was prepared in two steps. The base media (all components except human insulin and FBS) was first prepared. The base media was prepared in bulk, pH was adjusted to 6.7 using NaOH, it was sterile filtered, aliquoted into 50 ml tubes wrapped in foil, and then stored at –80 °C. This is because Grace's insect medium is unstable in solution and sensitive to light. It will begin to precipitate in 2–4 weeks when stored at 4 °C. To ensure the media was always fresh, a 50 ml aliquot of base media was kept at –80 °C until ready for use, thawed, and then used for no longer than 2 weeks (storing at 4 °C in the dark between use), after which it was discarded. This ensured the media was only subjected to one free-thaw cycle and guarded against precipitants forming in the media. Nonetheless, the media was checked for precipitants prior to each use and would not be used if any were found. For each experiment, 5–10 ml of complete media was prepared by drawing the necessary amount of base media and then adding human insulin and FBS immediately prior to use.

## Dissections

Imaginal eye discs were isolated from mid wandering third instar larvae (just as they began transitioning from the food onto the walls). No preference was given to males or females. Larvae were collected from vials and washed in water for up to 5 min at room temperature. They were sterilized in 70% (v/v) ethanol for 2 min and then briefly washed twice in 1 x PBS prior to transferring to 0.5 ml of complete culture medium in a glass well dissecting dish at room temperature. Larvae were left in culture medium for no longer than 30 min before dissection. Dissections were performed in a different well containing 0.5 ml of complete culture medium. No more than two to three larvae were dissected in a single well to prevent build up of debris. Additionally, care was taken to avoid rupturing the intestines during the dissection process in order to minimize contamination of the media. If the intestines were ruptured, discs isolated from that dissection well were discarded.

The cephalic complex (eye/antenna disc - mouth hook - optic lobe/ventral ganglion) was isolated by pulling the mouth hooks through the anterior end of the larva (although the 'inverted-sock' method could be used as well) using a pair of fine forceps (Dumont 0208–5-PO). The eye/antenna

disc was carefully dissected from the mouth hooks and optic lobes. The optic stalks of the eye discs were first separated from the optic lobes. The eye/antennal discs / mouth hooks were held by pinning the mouth hooks against the bottom of the dissection well using one pair of forceps, while the optic lobes were carefully pulled off the eye disc using the other pair of forceps. Next, the antenna disc was cut from the mouth hooks using a 26 G syringe (26G × 5/8, BD Precision Glide 305115), making sure to cut the tissue on the mouth hooks and leave the peripodial membrane of the antenna disc undamaged. If the peripodial membrane of the eye/antenna disc became damaged during the dissection process, it led to mechanical instability during the imaging. Thus, care was taken to prevent peripodial damage, and if it was observed, the sample was discarded. Eye/antenna discs remained more mechanically stable when left attached to the mouth hooks, but this precluded high resolution imaging because the mouth hooks are too bulky to allow the discs to lay flat against a cover slip.

## Tissue mounting

Mounting proved as important for successful ex vivo eye disc development as the media composition. Without appropriate mechanical support, the eye disc collapses basally and folds along the length of the morphogenetic furrow (MF), bringing the posterior and anterior compartments of the disc in physical proximity. Additionally, the antennal disc presses against the anterior compartment of the eye disc. Within 2–4 hr, the eye/antenna disc comes to resemble a folded piece of paper.

There are a variety of published methods for mounting imaginal discs for ex vivo imaging. Some methods involve embedding discs in agarose directly against a cover slip (*Tsao et al., 2016*), while others involve pinning the disc between the cover slip and a porous membrane (*Aldaz et al., 2010*; *Dye et al., 2017*). We hybridized these approaches to create a uniquely mechanically rigid environment for the disc to develop and be imaged.

We screened a variety of agarose types and concentrations in order to find one with optimal stiffness. If the stiffness of the agarose was not sufficiently high, eye discs would fold in the manner described above, as if they were not embedded in agarose at all. If the agarose was too stiff, the MF would stop moving after one to two hours. We discovered that SeaKem Gold Agarose (Lonza Rockland, #50150) used at 0.7% (w/v) concentration would reliably stabilize the eye/antenna disc while also permitting movement of the MF. Prior to the start of dissections, a 20 ml solution of 0.7% (w/v) SeaKem Gold Agarose in 1 x PBS was prepared in a small beaker by making molten in a microwave and then kept molten in a hot water bath set to 45°C with a piece of parafilm over the top of the beaker to prevent evaporation.

Eye/antenna discs were mounted in MatTek glass bottom petri dishes (35 mm dish, No. 1.5 coverslip, MatTek P35G-1.5–14 C). MatTek dishes were pre-coated with poly-L-lysine (Sigma P4707) to help adhere the eye discs against the cover slip during the mounting process. To pre-coat the dishes, the coverslips in the center of MatTek dishes were coated in ~0.5 ml of 0.01% (w/v) poly-L-Lysine and incubated at room temperature for 5 min before the solution was removed. They were dried at room temperature overnight and stored in the dark at room temperature for up to 2 weeks prior to use.

To construct a mounting/imaging chamber, a piece of double sided sticky tape (50 μm thick, Tesa Inc #05338) was cut into a ~ 15 x 15 mm square, whole punched (0.25 inch / 6 mm diameter) with a sterile hole puncher (Staples 10,573 CC), and placed flush against the cover slip in the bottom of a poly-L-Lysine coated MatTek petri dish (*Figure 1C*). A piece of Whatman Cyclopore polycarbonate membrane (5 μm pore size, Sigma WHA70602513) was cut to the same size as the piece of double-sided tape (~15 x 15 mm) and set aside.

After 10–15 eye/antenna discs were dissected, they were transferred to the hole-punched well in the MatTek dish. This was done by using a P20 pipet tip pre-lubricated with FBS and with the bottom ~2 mm cut off. Transfer usually resulted in ~300 μl of complete media being transferred along with the discs into the hole-punched well. The discs were arranged so they were non-overlapping and oriented with their apical surface facing the cover slip. The media that was transferred along with the discs was drained away using a micropipet, leaving the discs flattened against the cover slip with a trace amount of media preventing them from desiccating. Very quickly, 2.5 μl of the agarose solution was applied to the discs and allowed to set for 5 s. An additional 2.5 μl of agarose solution was then applied and allowed to set. The precut Whatman membrane was laid across the hole-punched well now filled with agarose, making sure no air bubbles were trapped

between the agarose and membrane. The edges of the Whatman membrane were pressed firmly against the tape to ensure formation of a strong bond. Finally, the MatTek dish was filled with 2 ml of complete media.

## Microscopy

Samples were imaged with a laser confocal microscope system at room temperature with z-stacks collected at precise 5 min intervals. We used Leica Sp5 (wildtype replicate 1), Zeiss LSM 800 with no airy scanner (wildtype replicate 2), and Leica Sp8 (*scabrous* strong and weak mutants) systems. Wildtype replicate one was imaged on a Leica Sp5 with a detector size of 1024 × 1,024 using the 100 x/1.44NA objective, 3 x internal zoom, and the pinhole set to one airy unit, resulting in a voxel size of 0.0505 × 0.0505 x 0.5003 µm and a xy field-of-view of 51.72 × 51.72 µm. The sample was scanned at 8000 Hz using the resonance scanner with bidirectional scanning and 16 x line averaging. A total depth of 29.71 µm was collected using 60 steps of 0.5 µm each. The E-cadherin-GFP reporter was excited using the 488 nm channel of the argon laser (set to 25%) at 15% and collected on a HyD-PMT (GaAsP) detector with 300% gain. Wildtype replicate two was was imaged on a Zeiss LSM 800 with a detector size of 634 × 634 using a 63 x/1.2NA objective, 2 x internal zoom, and the pinhole set to 1.29 airy units, resulting in a voxel size of 0.08 × 0.08 x 0.7 µm and a xy field-of-view of 50.71 × 50.71 µm. The sample was scanned at 1000 Hz using the point scanner with bidirectional scanning and 4 x line averaging. A total depth of 28.7 µm was collected in 42 steps of 0.683 µm each. The E-cadherin-GFP reporter was excited using the 488 nm laser at 0.8% and collected on a GaAsP-Pmt detector using 652 V of gain. Both *scabrous* datasets were collected on a Leica Sp8 with a detector size of 704 × 704 using a 63 x/1.4NA oil immersion objective, 2 x internal zoom, and the pinhole set to one airy unit, resulting in a voxel size of 0.0829 × 0.0829 x 0.4 µm and a xy field-of-view of 61.60 × 61.60 µm. A total depth of 9.2 µm was collected in 24 steps of 0.38 µm each. The E-cadherin-GFP reporter was excited using the Diode 488 nm laser at 0.8% and collected on a HyD-PMT (GaAsP) detector with 200% gain.

Note that the different wildtype replicates were imaged using different microscope systems and yet had highly reproducible behaviors for a wide spectrum of tissue and cellular features. Moreover, the *scabrous* mutant discs also exhibited highly reproducible behaviors similar to wildtype. Thus, different imaging platforms had little or no effect on the biological behaviors that we analyzed. All samples experienced photobleaching, evidenced by diminishing GFP signal over the course of imaging, and most likely experienced phototoxicity. We did not systematically test the effect of phototoxicity on the tissue. Generally, bleaching of the GFP reporter leveled off after about three or 4 hr of imaging. Laser powers were selected to generate sufficiently bright signal for segmentation after the point when GFP bleaching stabilized. It was not necessary to excite the GFP reporter up to the point of saturating pixel brightness, as is often done with fixed samples. Instead, we found that exciting the GFP reporter such that the signal was easily visible by eye was sufficient for cell segmentation.

Imaging resolution was selected to be sufficient for accurate cell segmentation within the morphogenetic furrow (MF). Cells in the MF are maximally constricted in their apical domains with the smallest cell diameters being around 0.5 µm. The imaging resolution of wildtype replicate one resulted in ~10 pixels along a 0.5 µm cell edge, whereas the imaging resolution of wildtype replicate two and both *scabrous* datasets led to ~6 pixels along a 0.5 µm edge. While we were able to segment wildtype replicate two and the *scabrous* datasets, wildtype replicate one was considerably easier to segment because of the higher resolution. We would advise future datasets to be collected at this resolution.

All samples were imaged about half-way between the margin of the eye disc and the equator. We did not specifically image dorsal only or ventral only regions. We found this region to be most mechanically stable. Often, the equatorial region of the disc would slowly sink basally over the course of imaging, requiring extremely large z-volumes to ensure the apical surface remained within the z-stack over the course of the movie. This resulted in excessive bleaching and diminishing signal intensity as the distance between the apical surface and cover slip increased. The average spacing of ommatidia changes as a function of dorsal-ventral (DV) position within the tissue, with tighter spacing near the equator and larger spacing near the margin. Imaging at the same relative DV position in discs from animals of approximately the same age was essential for keeping the spacing of ommatidia similar across datasets.

## Image processing

We created an image processing pipeline to segment and track all cells within a chosen field of view (*Figure 1—figure supplement 1*). This pipeline involved the following steps in this order: (1) image collection, (2) surface projection, (3) image registration, (4) cell segmentation, (5) cell tracking. These steps are explained in more detail below.

## Surface detection and projection

Cells were fluorescently labeled using an E-cadherin-GFP fusion protein that localizes to the adherens junction (AJ) surface of the eye disc epithelium. The E-cadherin-GFP protein also labels cells in the peripodial membrane, an epithelium of squamous cells located apical to the disc epithelium and separated from the disc epithelium by a fluid-filled lumen (*Figure 1—figure supplement 1*). In order to get a 2D projection of the AJ plane with signal-to-noise that permits downstream image analysis, and without inclusion of the peripodial membrane signal, we used the open-source MATLAB based software ImSAnE (Image Surface Analysis Environment) (*Heemskerk and Streichan, 2015*).

The first step in the surface detection workflow is the surfaceDetection module. We used the MIPDetector class of the planarDetector method of the surfaceDetection module. The MIPDetector and planarEdgeDetector classes are useful for finding approximately planar surfaces such as the AJ surface in imaginal discs. The MIPDetector relies on finding the brightest z-position for every xy coordinate in the gaussian derivative of the 3D image stack (i.e. after smoothing with a Gaussian filter). ImSAnE does this in a rescaled version of the 3D image stack where the pixels have been interpolated so that the physical voxels they represent have a unit aspect ratio of 1 (our image voxels were anisotropic with z-size ~10 x larger than xy size, see section above). Because cells are more densely packed within the AJ surface than in the peripodial membrane surface, the AJ surface creates a more continuous and brighter structure in the Gaussian filtered 3D stack, which allows for detection via the MIPDetector method. This method additionally relies on exclusion of outliers (with regard to z-position) by looking at the distribution of z-positions within local xy neighborhoods. Representative parameters for the MIPDetector are sigma 5, channels 1, zdir –3, maxIthresh 0.05, summedIthresh 0.5, sigZoutliers 1, scaleZoutliers 50, and seedDistance 20. For some time points, it proved impossible to get the MIPDetector to specifically recognize the AJ surface of the disc epithelium without inclusion of portions of the peripodial membrane. This would happen at xy coordinates where there were dark interior of large cells (often dividing cells) in the AJ surface aligned with bright portions of the peripodial membrane signal. For these time points, we found that manually masking the peripodial membrane signal using ImageJ allowed us to isolate the AJ surface in ImSAnE.

The next step in the surface detection workflow is the surfaceFitting module. We used the tpsFitter method of the surfaceFitter class of the surfaceFitting module. Briefly, the tpsFitter fits thin plate splines to the surface produced by the MIPDetector, which is stored as a 3D point cloud. This method only works for surfaces found using the MIPDetector or planarEdgeDetector classes of the planarDetector method, which have a single z point for every xy coordinate. We set the smoothing parameter to 1,000 and used a grid size of 80 × 80 for all datasets. The typical image size ranged from 600 × 600–1024 x 1024 pixels. This smaller grid size was chosen because it significantly sped up the spline fitting process and did not affect the quality of the fit. ImSAnE also allows the surface to be shifted in the z direction using the fitOptions shift method. We shifted the surface by ~10 pixels in the interpolated z-space, which corresponds to ~0.5 μm in physical space, in order to intersect the denser region of signal. The final step of the surfaceFitting module is creating a surface pullback using the pullbackStack method. We additionally used the onionOpts class of the pullbackStack method, which creates a 'z-stack' around the tpsFitter surface with a specified number of slices at a specified spacing in the interpolated z-stack space; we used 21 slices with a spacing of 1 for all datasets. The onionOpts class generates a max intensity projection of this 'z-stack' around the surface. This MIP was used as the final 2D projection of the image for segmentation and further analysis. Using onionOpts was integral to the surface projection workflow, as there are always regions where the surface isn't perfectly fit to the brightest portions of AJ signal and onionOpts helps capture the entirety of the signal in these regions.

ImSAnE also includes a surfaceAnalysis module with a Manifold2D class that provides a map between the 3D surface and 2D projection and tools for making distortion free measurements of area and paths along the surface. However, we did not use this module of ImSAnE for our analysis because the image registration we used to handle tissue drift (see below) created a new reference frame for the

segmented images that is distinct from the reference frame of surface projection. We tested the error resulting from distance measurements in the 2D projection without regard for the 3D curvature of the surface vs. distance measurements made using the surfaceAnalysis module. Curvature is greatest along the AP axis, with little curvature along the dorsal-ventral (DV) axis. For distance measurements around the length scale separating cell centroids (~1–2 μm) along the anterior-posterior (AP) axis, which is also about the average velocity cells move within the morphogenetic furrow (MF) over 1 hr, we found that disregarding 3D curvature introduced a ~ 1.5% error compared to making the same measurements with ImSAnE's surfaceAnalysis module (data not shown). Measurements made in the DV direction, where there is very little curvature, had only a ~ 0.1% error. The curvature of the surface is greatest in the AP direction around the location of the MF, where we observe periodic cell flows in the anterior direction. Therefore, this error is affecting our measurements of the velocity of all cells in the MF region and the magnitude of these periodic cell flows. However, this measurement error is not affecting our analysis of the DV organization of these cell flows because there is very little measurement error along that axis.

## Image registration and alignment of the AP/DV axes

After creating 2D surface projections using ImSAnE, we next eliminated tissue drift using image registration. The ex vivo tissue had a tendency to drift in the anterior direction while imaging. This was often most severe toward the start of the ex vivo culture. The drift was sometimes so severe that it required adjustment of the imaging field-of-view over the course of imaging, such as with wild-type replicate 1. This was to prevent the region-of-interest near the morphogenetic furrow (MF) from drifting out the field-of-view. In order to eliminate the effect of tissue drift from our analysis of cell movement, we tested a number of different intensity and feature based registration algorithms in MATLAB and ImageJ and found that the Linear stack alignment with SIFT plugin for FIJI (*Schindelin et al., 2012*) produced the most stable movie – that is the smoothest movie with the least amount of random jittering between consecutive time points. The Linear stack alignment with SIFT plugin in FIJI is a JAVA implementation of the Scale-Invariant Feature Transform method (*Lowe, 2004*). Prior to registration, we padded the images with zeros in order to create a buffer region so that no portion of the image would be translated out of the field-of-view during the registration process. We registered together consecutive time points chronologically in time using a Rigid transformation and the following parameters. For the Scale Invariant Interest Point Detector, we used a 1.6 pixel kernel for the gaussian blur, 3 steps per octave, 64 pixel minimum image size, and 1024 pixel maximum image size. For the Feature Descriptor, we used a feature descriptor size of 4, 8 feature descriptor orientation bins, and a closest/next closest ratio of 0.92. For the Geometric Consensus Filter, we used a maximum alignment error of 10 pixels and an inlier ratio of 0.05.

After removing tissue drift with image registration, the movie was translated such that the MF was parallel to the y-axis of the image. Since the MF is also parallel to the dorsal-ventral (DV) axis, this ensure that the DV and y axes were aligned, and the anterior-posterior and x axes were also aligned.

## Segmentation and cell tracking

After creating 2D surface projections using ImSAnE and eliminating tissue drift with image registration, the final pre-processing steps before analysis were cell segmentation and tracking. Cell segmentation was achieved using a combination of a convolutional neural network (CNN) for pixel classification and MATLAB scripts for cell detection and tracking. Cell tracking relied on a MATLAB implementation of the Munkres assignment algorithm (*Cao, 2021*; *Kuhn, 1955*). Our code is publicly available (https://github.com/K-D-Gallagher/eye-patterning, copy archived at swh:1:rev:fef5ba1e4fa90f01b-55b67c74bf429c6c873acc1; *Gallagher, 2021a*).

We trained a CNN to classify pixels as either cell edges or background/cell interior (https://github.com/K-D-Gallagher/CNN-pixel-classification, copy archived at swh:1:rev:33f9a07bd8dac-b3a2554a1788dd9b60b267162d4; *Gallagher, 2021b*) (*Figure 1—figure supplement 1*). We did this using a Pytorch package that provides transfer learning via pre-trained encoders paired with a variety of untrained decoder architectures that can be learned towards your specific pixel classification task (Yakubovskiy, 2020). This allowed us easily explore a variety of CNN architectures and find the most accurate one for our data. We used a watershed transform to detect cells from the CNN pixel classification output and determined that our CNN was capable of accurately

segmenting ~99.5% of cells relative to manually curated grown truth dataset (see below). However, when we attempted to track cells in datasets with ~0.5% error in cell segmentation, we were only able to accurately track ~80% of cells relative to our growth truth data. Errors in cell segmentation compound geometrically into errors in cell tracking. Therefore, we developed a custom MATLAB software to manually correct errors in segmentation. This software uses errors in cell tracking to discover the underlying errors in segmentation. Errors in cell segmentation were corrected using this software until no further errors in cell tracking could be detected. After manually correcting cell segmentation, ~ 35% of segmented cells could be tracked through the entire course of the movie. The remainder of tracked cells either appeared/disappeared off the boundary of the field-of-view (FOV) (~55% of segmented cells) or appeared/disappeared within the interior of the FOV (~10% of segmented cells). This latter category we surmised to be cell birth and death events (see below). In sum, we could account for the behavior of 100% of segmented cells as either (1) being tracked throughout the entire duration of the movie (~35% of all segmented cells), (2) biological birth/death events (~6 % / ~4% of all segmented cells), or (3) appearing/disappearing over a boundary of the FOV (~55% of all segmented cells).

There are no publicly available datasets for training CNNs to segment fluorescently labeled epithelial cells. Therefore, to train our CNN, we generated our own ground truth training set. This was done using Ilastik, an open-source machine learning (random forest classifier) based image processing software that offers segmentation, classification, and tracking workflows (*Berg et al., 2019*). Mistakes remaining after pixel classification in Ilastik were hand corrected using the above-mentioned MATLAB software. Relative to this hand corrected ground truth dataset, Ilastik was capable of accurately segmenting ~93% of cells (compared to ~99.5% with our CNN). The initial manually corrected ground truth training dataset for our CNN was approximately 120 images (one wildtype replicate) but progressively grew as new datasets were segmented and manually corrected. We also obtained confocal data of the *Drosophila* wing imaginal disc and thorax generously shared by Ken Irvine and Yohanns Bellaïche, respectively. Including this data in our training data not only increased the training library volume, which increases CNN model accuracy, but also increased the variability in the training dataset, which confers greater generalizability to new datasets not represented in the training data. Our trained CNN model is publicly available (https://github.com/K-D-Gallagher/CNN-pixel-classification).

The entire FOV was not segmented in order to reduce the necessary amount of manual correction of segmentation errors, as this was the most rate limiting step in our pipeline. We only segmented ~20 μm±the location of the morphogenetic furrow (MF), as this was our region-of-interest for analysis. Because the MF moves towards the anterior edge of the FOV, the number of segmented cells anterior to the MF progressively diminishes over the course of the movie. For example, the segmented FOV of wildtype replicate one begins with with 732 cells (compared to ~1000 cells in the total image, including non-segmented cells) and ends with 458 segmented cells at the last frame of the movie; the reduced number of segmented cells is primarily to the anterior of the MF.

## Morphogenetic furrow detection

The morphogenetic furrow (MF) was defined as a 1D position along the anterior-posterior (AP) axis. Its location was determined by the maximum peak in the DV average of E-cadherin-GFP signal along the AP axis. 1D profiles of cell density for every pixel along the AP axis were defined for each time point by summing the raw E-cadherin-GFP signal after diffusing the image with a Gaussian filter with a sigma of 10 pixels. This resulted in a 1D cell density profile along the AP axis that contained a series of peaks that mapped to different features of the disc. The most anterior peak corresponded to the MF and then a series of smaller peaks to the posterior corresponded to the columns of ommatidia. By selecting the most anterior peak in 1D cell density profile, we were able to reliably find the MF in ~90% of time points. The errors in ~10% of time points resulted from the high density of E-cadherin-GFP signal in the arcs just posterior of the MF. This resulted in merging of the density peak of MF with the first column of ommatidia, leading to one broad peak in density that prevented accurate determination of the exact location of the MF. For these time points, the position of the MF was manually assigned. After determined the location of the MF for each time point, the vector describing MF position over time was smoothed by averaging adjacent time points.

## Cell birth and death detection

With manual correction of cell segmentation and tracking ensuring 100% tracking accuracy, cell birth and deaths were identified using tracking information. Excluding tracks that began/ended along the boundary of the segmented field-of-view, cell births were identified by tracks that begin later than the first time point and cell deaths were identified by tracks that ended prior to the last time point. We observed the morphological hallmarks of cell division corresponding to all identified instances of cell birth in the wildtype datasets (cells rounding followed by cytokinesis, *Figure 1G*). In the *scabrous* datasets, there were instances of cell division that had little cell rounding before cytokinesis, which we believe is a consequence of the increased cell density in the interommatidial space to the posterior of the morphogenetic furrow in these mutants. In both the wildtype and *scabrous* datasets, we observed the morphological attributes of cell delamination corresponding to all instances of cell death (*Figure 1—figure supplement 2D*). Cells progressively restricted their apical cross-sectional area prior to disappearing and then never reappeared.

## Velocity and area change measurements

Velocity was calculated as the change in xy position of a cell centroid over 12 frames of the movie (1 hr real time) after the tissue drift had been eliminated via image registration (see above). Cell velocity was calculated using the central difference of the centroid position – that is the change in cell centroid position from t-6 to t + 6. Therefore, the velocity of each cell was measured from the 7th time point that cell was tracked until seven time points before that cell track ended. Because the anterior-posterior (AP) axis was aligned to the x-axis and the dorsal-ventral (DV) axis aligned to the y-axis, the x-component of a velocity vector described the movement of that cell along the AP axis, whereas the y-component described its movement along the DV axis. The rate of area change was calculated by the difference in the number of pixels within a segmented cell over 12 frames of the movie, also using the central difference. Similar to velocity, the rate of area change of each cell was measured from the 7th time point that cell was tracked until seven time points before that cell track ended.

Cell velocity and rate of area change was measured over 12 frames (1 hr real time) as this was found to be the timescale over which deterministic behavior emerged from noise. In other words, we tried calculating velocity with time steps ranging from one frame (5 min real time) to 24 frames (2 hr real time) and found that the behavior of cells started to stabilize with a time step of 12 frames. Using time steps less than 12 movie frames resulted in a noisier velocity field with less clear features and organization, whereas including more than 12 frames did not substantially change the qualitative appearance / organization of the velocity field compared to that derived from a tilmestep of 12 frames. Twelve frames was found to be the deterministic time scale for the rate of area change measurement as well.

## Odd/even velocity profiles

The position of odd and even rows was manually defined by inspecting the morphology of each dataset. The average anterior-posterior (AP) velocity of cells within each odd and even row at the position of the morphogenetic furrow (MF) was calculated as follows. Search windows 2.5 × 2.5 μm were positioned at the dorsal-ventral (DV) coordinate of each odd and even row and the AP coordinate of the MF. Over the course of the movie, the search windows were moved along the AP axis according to the position of the MF. The mean AP velocity of cells within each odd/even search window at a given time point was calculated and the 95% confidence interval of the mean was bootstrapped. Measurements from cells contained within the different odd-row search windows were all averaged together and the same was done for the cells contained within different even-row search windows. For the autocorrelation analysis, cells contained within different odd or even search windows were not averaged together within the same time point. Instead, separate profiles over time were created for each odd and even search window. These profiles were then averaged and their 95% confident intervals were bootstrapped.

## 1D moving window profiles

The 1D profiles of velocity, area, and rate of area change were collected from time points 7 (35 min) to 60 (5 hr) for each dataset. Starting at time point seven is due to the way velocity and rate of area change are calculated (see above). Time points after 5 hr were excluded in order to avoid potential artifacts from the reduced rate of morphogenetic furrow (MF) movement and proliferation seen the

later parts of ex vivo cultures. The mean and 95% confidence interval of cell velocity, area, and rate of area change were calculated for every position along the anterior-posterior (AP) axis using a sliding window of ~0.5 µm width. Cells with centroids within the 0.5 µm sliding window search area were included in the calculation for each position along the AP axis. Measurements were collected independently for all time points and then aligned according to the position of the MF. After alignment of all the time points, the mean and 95% confidence interval were computed for each position along the AP axis.

### Distribution analysis

Distribution measurements of AP velocity were collected in a similar manner to the 1D profiles, described above. However, instead of collecting measurements at every position along the anterior-posterior (AP) axis using a sliding window, measurements were collected using bins of defined width and positioned relative to the location of the morphogenetic furrow (MF). There are six bins spanning the anterior-posterior (AP) axis. The following are the boundaries of the bins, relative to the position of the MF (0 µm), where the anterior compartment is positive and the posterior compartment negative: 15 µm, 7.5 µm, 2.5 µm, –2.5 µm, –7.5 µm, –12.5 µm, and –20 µm. The 'anterior MF' bin spans 2.5 µm to –2.5 µm, while the 'posterior MF' bin spans –2.5 µm to –7.5 µm. The 'posterior transition zone' spans –7.5 µm to –12.5 µm and the 'far posterior' spans –12.5 µm to –20 µm. Similar to the 1D profiles, any cell with a centroid within the bin search range at a given time point was included in that distribution.

### Fourier transforms of AP velocity

Fourier transforms (FTs) were calculated using the same binning strategy as the velocity, area, and rate of area change distributions, described above. Briefly, measurements were collected in six bins along the anterior-posterior (AP) axis; these bins had fixed width and were positioned according to the location of the morphogenetic furrow (MF) at each time point. Measurements were collected from time point 7–60 for the same reasons outlined above.

The centroid based velocity field is irregularly spaced. Prior to calculating the FTs, the AP component of the velocity field was interpolated onto a regularly spaced grid using a scattered interplant routine built in to MATLAB. Interpolated grid points were spaced 1 µm apart in both the AP and dorsal-ventral (DV) direction. The FT of the AP velocity was computed using the built in Fast Fourier Transform routine in MATLAB. FTs were calculated independently for every row of interpolated AP velocities (where a row is defined as a single position along the AP axis, extending in the DV direction). In order to set the zero mode in the frequency domain to zero, we subtracted the mean AP velocity of a given row from the interpolated values of that row prior to computing the FT. The zero mode was set to zero in order to gain better resolution in higher frequency signal that would have otherwise been dwarfed by the zero mode peak. The FT for each row was assigned to one of the six bins along the AP axis according to its position and the FTs were averaged together after computing all FTs for a given bin across all time points.

After setting the zero mode to zero, there remained significant density in the next lowest frequencies near the origin. This is due to long wavelength modulations along the DV axis. This is likely resulting, in part, from a gradient in ommatidia maturation that runs along the DV axis. For every DV column of ommatidia, those closest to the equator mature more rapidly than those near the margin (*Wolff and Ready, 1993*). This creates a V shaped pattern of ommatidial maturation within each column of the ommatidial lattice and creates a velocity gradient along the DV axis.

### R cell E-cadherin-GFP profiles

Cellular E-cadherin-GFP concentration was calculated as the average E-cadherin-GFP intensity within each segmented cell. Because segmentation is not perfect and membrane bound E-cadherin-GFP could belong to either cell that shares a given membrane, masks of segmented cell area were dilated by two pixels to include signal underneath and on the other side of the segmentation skeleton prior to calculating averaged E-cadherin-GFP levels. This means that there is a minor amount of E-cadherin-GFP signal that is getting counted towards multiple cells whenever R cells are touching. While bleaching was present in all datasets, it was not corrected for in this analysis because the data collapsed cleanly once the trajectories of individual ommatidia were aligned. Trajectories of individual ommatidia were aligned based on morphological features intrinsic to each ommatidia;

specifically, ommatidia were aligned based on the when both R2 and R5 as well as R3 and R4 first touch. This morphological feature marks the moment that arcs close into 5-cell rosettes (*Figure 1F*) and synchronizes the behavior of ommatidia that exited the morphogenetic furrow (MF) at different times, as well as synchronizes ommatidia across multiple datasets. Ommatidia that either never meet this condition or already met this condition at the start of the movie were omitted from this analysis. A cohort of non-R cells were chosen for every ommatidia included in this analysis by selecting non-R cells touching any one of the five R-cells at the time point when the 5-cell rosette closed. This ensured that the non-R cells were at approximately the same distance from the MF as the R-cells throughout their sampling.

## Acknowledgements

We wish to thank Henry Sun, Nathalie Dye and Suzanne Eaton for kindly sharing their growth media recipes before publication. We thank Nathan Burg and Robert Gray for help in processing some of the movies and CNN training. We thank Nicolas Pelaez for his advice on live eye imaging. We thank Ken Irvine and Yohanns Bellaïche for sharing their data assist in developing the CNN pixel classification model. We want to thank the Kavli Institute of Theoretical Physics at UCSB (NSF Grant PHY-1748958) for hosting the authors and providing stimulating discussions about the work. In particular we thank Boris Shraiman at KITP for discussions. We thank Jessica Hornick and the Biological Imaging Facility at Northwestern (RRID:SCR_017767). Financial support was provided from the NIH (R35GM118144, RWC), NSF (1764421, MM and RWC), and the Simons Foundation (597491, MM and RWC). MM is a Simons Foundation Investigator. KDG is a NSF Simons Center Scholar.

## Additional information

### Funding

| Funder | Grant reference number | Author |
| --- | --- | --- |
| National Institutes of Health | R35GM118144 | Richard W Carthew |
| National Science Foundation | 1764421 | Madhav Mani<br>Richard W Carthew |
| Simons Foundation | 597491 | Madhav Mani<br>Richard W Carthew |

The funders had no role in study design, data collection and interpretation, or the decision to submit the work for publication.

### Author contributions

Kevin D Gallagher, Data curation, Formal analysis, Investigation, Methodology, Resources, Software, Writing - review and editing; Madhav Mani, Conceptualization, Formal analysis, Methodology, Project administration, Supervision, Writing - review and editing; Richard W Carthew, Conceptualization, Formal analysis, Funding acquisition, Project administration, Supervision, Writing - original draft

### Author ORCIDs

Richard W Carthew ⬤ http://orcid.org/0000-0003-0343-0156

### Decision letter and Author response

Decision letter https://doi.org/10.7554/eLife.72806.sa1
Author response https://doi.org/10.7554/eLife.72806.sa2

## Additional files

### Supplementary files
• Transparent reporting form

## Data availability

All imaging data has been deposited in Dryad (doi:http://doi.org/10.5061/dryad.f4qrfj6wp).

The following dataset was generated:

| Author(s) | Year | Dataset title | Dataset URL | Database and Identifier |
|---|---|---|---|---|
| Carthew RW, Gallagher K, Mani M | 2021 | Emergence of a geometric pattern of cell fates from tissue-scale mechanics in the Drosophila eye | http://dx.doi.org/10.5061/dryad.f4qrfj6wp | Dryad Digital Repository, 10.5061/dryad.f4qrfj6wp |

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
