## [Editor Report]

This paper will have a high impact for biologists who are interested in tissue patterning and organogenesis. It provides unexpected insights into the problem of regular spacing of sub-organ structures. The study is based on innovative live imaging technology with state of the art analysis tools.

---

## [Decision Letter]

**Decision letter after peer review:**

Thank you for submitting your article "Emergence of a geometric pattern of cell fates from tissue-scale mechanics in the *Drosophila* eye" for consideration by *eLife*. Your article has been reviewed by 2 peer reviewers, and the evaluation has been overseen by a Reviewing Editor and Utpal Banerjee as the Senior Editor. The reviewers have opted to remain anonymous.

The reviewers have discussed their reviews with one another, and the Reviewing Editor has drafted this to help you prepare a revised submission. The reviews are obviously positive and laudatory. Congratulations! I suggest that you either do the experiments proposed by reviewer 2 using the Rock inhibitor or make textual changes to accommodate the issues related to cause and effect. the remainder are small changes.

*Reviewer #1 (Recommendations for the authors):*

The observation that the regular ommatidial spacing is mechanically driven is ground breaking and convincing. The authors propose that the regulated, evenly spaced flow of cells (from posterior to anterior) is depending on two features: (i) the dilation zone in the back that creates pressure, and (ii) the function of Scabrous (Sca), which has been shown to be critical for the regular spacing. As the spacing requirement of Sca is well documented it is not surprising that Sca is required. The function of Sca – in this new mechanical model – is however unexpected. As the authors note Sca has been linked to lateral inhibition and reaction-diffusion based spacing models, as it also has been proposed to potentially act as a ligand for Notch (which never made much sense…). Sca encodes a secreted Fibrinogen-related molecule and so its new proposed function, to modulate the basal membrane attachments and thus allow or inhibit the cell flow, intriguing. In that context it is noted that the "flow" shows variability between strong and weak sca mutant discs (these mutants apparently show quite a bit of variability, which is to be expected for such a 'function'). The idea how Sca really modulates the flow levels and the flow periodicity is however somewhat lost in the text. While the paper is very well written overall, it would help a lot to be very specific what the authors think Sca is doing or not doing. As a secreted Fibrinogen-related factor does it modulate the extra-cellular matrix, or does it act at the level of the basement membrane attachments to better (or worse) attach the cells surrounding an R8-centered precluster?

In this context two previously published studies on Sca should be discussed, and the respective results might help in further defining how Sca is functioning in the "mechanical spacing model". While the loss-of-function (LOF) Sca mutants are critical (and analyzed in detail), what about the gain-of-function scenario? Ellis et al. (Development 1994) have suggested that Sca overexpression/misexpression displayed a phenotype very similar to its LOF mutant alleles. How would this fit within the new model?

Similarly, Sca has been linked to a potential function in ommatidial rotation (Chou and Chien, Dev Cell 2002) albeit through an unclear mechanism. Can the new mechanical model help to understand that, or does that data further instruct how Sca might affect the "cell flow"? Can the proposed function for Sca, if any (?), be consistent with these observations? Does Sca stimulate cell dilation?

Figure 1E: It is unfortunate that these panels shown do not display any dividing cells, although they are quantified in 1G and H. Similarly, it would be nice to also show a micrograph displaying cell delamination (as again it is quantified).

The arrangement of figure panels 2A vs 2B, and similarly 3A vs 3B are rotated 90 degrees respective to each other. In all other figure panels the A-P axis is aligned horizontally, and it would be easier to follow the figures if all panels are arranged in the same manner.

The expression levels of Ecad in R-cells are quantified. It would be nice to show some actual micrographs reflecting that.

Figure panels 5B and C are redundant with panels 5E and F, as these show the exact same curves for wild-type (and compared to sca mutants in the latter).

The use of statistical analyses appears lost. I might have missed it, but it is not clear from the figure legends for example what type of statistical analyses were applied, what the n and p values are etc.

Please indicate which sca LOF allele is being employed in the main text.

*Reviewer #2 (Recommendations for the authors):*

1. In order to claim that spatially-defined cell movement dictates the subsequent photoreceptor spacing, it is necessary to compromise cell movement without perturbing signaling or cell communication, and ask if the determination of R8 spacing was indeed altered or abolished. Since it is not known which cytoskeletal elements mediate the cell dilation that drives movement, it is not obvious what to perturb.

It is likely that actomyosin is involved. Since this is an ex vivo system, utilization of the Rock inhibitor is feasible. Blocking actomyosin activity and cell movement would then allow to test if the cell movement was indeed blocked, and if specification of cell fate still takes place.

Another option to perturb cell shape changes may be inspired by the Krueger et al. 2018 EMBO J. paper, where optogenetic activation of basal myosin prevented basal cell relaxation and delayed cell invagination during gastrulation.

These experiments may be beyond the scope of the current paper. However, without these results, it is only possible to say that cell movements correlate with photoreceptor spacing, rather than claiming that they are the source of the resulting pattern.

2. The major argument for the causal role of movement is the analysis of the scabarous mutant phenotype. This mutant background is complicated since there are claims that Scabarous is either a positive or a negative non-autonomous signal for photoreceptor specification. In addition, the effects of scabarous mutant phenotype are opposite immediately at the morphogenetic front (where it leads to a reduction in photoreceptor clusters), and several rows away (where excess clusters are observed due to catastrophes in periodicity of lateral inhibition patterns).

The authors observed a partial reduction in cell dilation in the scabarous mutant background. While alterations in cell migration in scabrous mutant discs were scored, the authors cannot determine if these alterations are the cause for the spacing disruption or a consequence of aberrant signaling. In short, since the scabarous data is incomplete and inconclusive, it would be better to present it as an analysis of a key mutant with the new understanding of cell movement in mind, and discuss the observations without using them to try and prove the model.

3. It would be helpful to relate the constant speed of furrow progression (formation of a new photoreceptor row every ~2.5 hours, driven by Hh and Dpp signaling), to the maximal migration speed of the cell clusters that do not differentiate (1.2-1.5 microns/hr). Are these cell clusters progressing with the furrow at the same speed?

4. In discussing their data, the authors focus strongly on the mechanism by which a periodic pattern of ommatidia is generated, but their interpretations actually constitute a considerably broader understanding and revision of the established developmental program for the differentiating eye imaginal disc. This "global picture", with its complexities, should be summarized in an encompassing model, which points out the key elements of their proposal in a single schematic (posterior and anterior "pressure zones", various cell flow patterns outside MF, transient immobility of ommatidial clusters, flow of clusters within the MF and their periodicity, onset of atonal expression, etc), to provide readers with a guide for piecing together a rather complex story.

Perhaps it is possible to consider a hybrid model for cell spacing, combining signaling and cell movement. In this model, cell migration would differentially impinge on the signaling distance between cells, but signaling would still be the initial cue. For example, cell clusters with reduced migration become more susceptible to inhibition by the older cluster, while the faster migrating cells become less susceptible and thus more prone to differentiate when the next photoreceptor row forms. A similar spacing argument was made for the potential role of constriction at the morphogenetic furrow, allowing lateral inhibition to reach more cells.

---

## [Author Response]

Reviewer #1 (Recommendations for the authors):The observation that the regular ommatidial spacing is mechanically driven is ground breaking and convincing. The authors propose that the regulated, evenly spaced flow of cells (from posterior to anterior) is depending on two features: (i) the dilation zone in the back that creates pressure, and (ii) the function of Scabrous (Sca), which has been shown to be critical for the regular spacing. As the spacing requirement of Sca is well documented it is not surprising that Sca is required. The function of Sca – in this new mechanical model – is however unexpected. As the authors note Sca has been linked to lateral inhibition and reaction-diffusion based spacing models, as it also has been proposed to potentially act as a ligand for Notch (which never made much sense…). Sca encodes a secreted Fibrinogen-related molecule and so its new proposed function, to modulate the basal membrane attachments and thus allow or inhibit the cell flow, intriguing. In that context it is noted that the "flow" shows variability between strong and weak sca mutant discs (these mutants apparently show quite a bit of variability, which is to be expected for such a 'function'). The idea how Sca really modulates the flow levels and the flow periodicity is however somewhat lost in the text. While the paper is very well written overall, it would help a lot to be very specific what the authors think Sca is doing or not doing. As a secreted Fibrinogen-related factor does it modulate the extra-cellular matrix, or does it act at the level of the basement membrane attachments to better (or worse) attach the cells surrounding an R8-centered precluster?

In the Discussion, we described our thoughts as to specifically what Sca might be doing (lines 428-435). We found that the scabrous mutant had both lower cell flow and a diminished cell dilation behind the MF (Figure 5). Thus, the most parsimonious explanation for why the mutant has lower flow is because a pressure gradient that is moving cells is weakened. It is formally possible that Sca also promotes flow by weakening attachments between flowing cells with one another or with the ECM, but this would necessitate two mechanisms Sca would control: pressure and attachment. We have reworded text in the Discussion to provide more clarity.

In this context two previously published studies on Sca should be discussed, and the respective results might help in further defining how Sca is functioning in the "mechanical spacing model". While the loss-of-function (LOF) Sca mutants are critical (and analyzed in detail), what about the gain-of-function scenario? Ellis et al. (Development 1994) have suggested that Sca overexpression/misexpression displayed a phenotype very similar to its LOF mutant alleles. How would this fit within the new model?

The Ellis paper does a GOF experiment in which they express Sca in other cells within each precluster besides R8 by using the rough enhancer. This still generates latticed expression of Sca but it perdures longer in developing ommatidia. Notably, the patterning of ommatidia as they emerge from the MF is not nearly as disrupted as what is seen in the LOF condition (Figure 3D,3F,3H in Ellis et al. 1994). However there is a strong disruption of later developmental events (Figure 2C,D,E,F). So this GOF outcome is not the same as the early patterning defects seen in the LOF condition. We interpret this to suggest that simply making more Sca in other R cells still generates the normal dilation of non R cells behind the MF because the early lattice pattern of Sca expression is not really altered.

Similarly, Sca has been linked to a potential function in ommatidial rotation (Chou and Chien, Dev Cell 2002) albeit through an unclear mechanism. Can the new mechanical model help to understand that, or does that data further instruct how Sca might affect the "cell flow"? Can the proposed function for Sca, if any (?), be consistent with these observations?

Chou and Chien observed that the same allele of scabrous we used has a small but significant over-rotation phenotype (Figure 5A in their paper). The mechanism behind this was not elucidated. Their observation does not really shed light on how Sca might stimulate the cell flows we observed. It is possible the Sca-induced posterior cell flow somehow brakes rotation though it is not easy to imagine how. We have added a short discussion of this connection in the Discussion.

Does Sca stimulate cell dilation?

Yes it does, and we showed it in Figure 5.

Figure 1E: It is unfortunate that these panels shown do not display any dividing cells, although they are quantified in 1G and H. Similarly, it would be nice to also show a micrograph displaying cell delamination (as again it is quantified).

Actually we show two sets of micrographic images of a dividing cell in figure panel 1G. We show images of a delaminating cell in Figure 1 – supplement 2D.

The arrangement of figure panels 2A vs 2B, and similarly 3A vs 3B are rotated 90 degrees respective to each other. In all other figure panels the A-P axis is aligned horizontally, and it would be easier to follow the figures if all panels are arranged in the same manner.

We certainly wanted to keep the axes in panels the same orientation, and we did so except for Figures 2B and 3B. The reason for making these exceptions was because the images are meant to provide a reader with a sense of where the various bins are located in Figures 2C and 3C. The color coding of cells is meant to visualize where the bin is located relative to the coupled charts in 2C and 3C. If we had placed panels 2B and 3B in a standard orientation, the coupled charts in 2C and 3C would have to be rotated 90 degrees. This would make their visualization unwieldy.

The expression levels of Ecad in R-cells are quantified. It would be nice to show some actual micrographs reflecting that.

Figure 1I does show a micrograph at low magnification where E cadherin signals are clearly stronger in R cell clusters. But in response to the reviewer’s suggestion, we have added a high magnification image in Figure —figure supplement 2A’ to make it clearer. Both are now cited in the relevant place in the results.

Figure panels 5B and C are redundant with panels 5E and F, as these show the exact same curves for wild-type (and compared to sca mutants in the latter).

This is formally correct in that the same data was analyzed. The reason we first show only the wildtype data (in 5B and C) is to describe the normal flow velocity and dilation profiles in 1D. The patterns are complex and not straightforward to describe clearly. The simplicity of a single line in each panel makes it easier for readers to grasp the complex behaviors. Our concern was that if we tried to describe the wildtype behavior in profiles that also included the mutant, readers would become lost in first comprehending the normal behavior without trying to compare normal and mutant. This was something we experienced when trying to use these plots (5E and 5F) to explain to people not involved in the work.

The use of statistical analyses appears lost. I might have missed it, but it is not clear from the figure legends for example what type of statistical analyses were applied, what the n and p values are etc.

Figures 2,3,5 and 6 have analyses in which the line averages are shown surrounded by shaded areas that are the 95% confidence intervals for the estimated line averages. That is most easy to see in Figure 3H. But all of the other charts have 95% CI shading also. The shaded areas are so narrow they are practically invisible relative to the width of the line average. If the reviewer zooms in as much as possible, they will see them. The reason the CIs are so small is the sheer number of cells sampled plus the remarkably consistent behaviors of the cells as a function of AP and DV position. For the cell distributions in Figure 3DEF, Figure 4DH, Figure 6B, we have performed non parametric Mann-Whitney U tests since the distributions are not all Gaussian. The results of those statistical tests are now presented in Table 1. The large sample sizes going into all our measurements resulted in extremely low p-values for many comparisons (i.e. p-values well below 0.05 for most comparisons, which was our reason for not including these statistics). Therefore, we report the -log10(p-values) of the Mann-Whitney U tests in order to allow readers to make more meaningful comparisons. The -log10 values report the statistical magnitude of the difference between one group and another; the smaller that value, the smaller the difference. For example, any -log10 value less than 1.3 means p > 0.05. All of the summary statistics are presented in a new Table 1. The older Table 1 is now Table 2.

Please indicate which sca LOF allele is being employed in the main text.

This has been done.

Reviewer #2 (Recommendations for the authors):1. In order to claim that spatially-defined cell movement dictates the subsequent photoreceptor spacing, it is necessary to compromise cell movement without perturbing signaling or cell communication, and ask if the determination of R8 spacing was indeed altered or abolished. Since it is not known which cytoskeletal elements mediate the cell dilation that drives movement, it is not obvious what to perturb.It is likely that actomyosin is involved. Since this is an ex vivo system, utilization of the Rock inhibitor is feasible. Blocking actomyosin activity and cell movement would then allow to test if the cell movement was indeed blocked, and if specification of cell fate still takes place.Another option to perturb cell shape changes may be inspired by the Krueger et al. 2018 EMBO J. paper, where optogenetic activation of basal myosin prevented basal cell relaxation and delayed cell invagination during gastrulation.These experiments may be beyond the scope of the current paper. However, without these results, it is only possible to say that cell movements correlate with photoreceptor spacing, rather than claiming that they are the source of the resulting pattern.

We thank the reviewer for their thoughtful remarks. The reviewer suggests we perturb the actomyosin system, assuming this would change the pressure gradient. However, adding a small molecule inhibitor of Rok to the imaging chamber would result in global disruption of actomyosin dynamics in all cells in the eye antennal disc complex. This would obliterate the MF as a structure, the myosin cages surrounding each precluster that keep them constricted (Escudero et al. 2007 Dev Cell 13, 717), and likely impact large scale interactions of the eye disc with the antennal disc and peripodial membrane. Such a catastrophic treatment will not yield meaningful results. The idea of using optogenetics is valid, focusing ectopic activation of myosin on non-R cells at the PTZ and observing the consequences. The technical aspects of such an experiment in a new ex vivo model other than the embryo make it years away from fruition and beyond the scope of this paper.

Moreover, we do not entirely agree with the reviewer’s claim that perturbing actomyosin dynamics will not perturb the underlying signaling and cell communication that is happening, and therefore provide a more rigorous means to test the model. Cell-cell signaling is coupled to cell trafficking and membrane dynamics (i.e. Moujaber and Stochaj 2020 Trends Biochem Sci 45, 96). F actin dynamics at the cell cortex profoundly influence exocytosis and endocytosis of materials from the surface. Actin dynamics have been shown to impact Hedgehog and Hippo signaling in the *Drosophila* eye disc (Benlali et al., 2000 Cell 101, 271; Bras-Perera et al. 2011 Com Int Biol 5, 612). If we do such an experiment to alter cytoskeletal dynamics, how will one be able to rigorously exclude that any result is due to altered cell-cell signaling of some form.

In essence, the Scabrous mutation leads to a diminished cell dilation in the PTZ, diminished cell flow, and broken emergence of the lattice at the MF. One scenario is that it directly acts in parallel on each of those processes, which are independent of one another (correlated without causal connection). It is difficult to imagine how a single protein could independently elicit such diverse cell behaviors as differentiation and cell size. A more likely scenario is that Scabrous directly acts on one of these processes and indirectly on the others – and the one directly regulated process regulates the others (correlated with causal connection). If so, there are two possible pathways. One, the broken lattice is upstream of the defects in pressure and flow, meaning the defects in spacing directly caused by the mutant have consequences on pressure and flow more posterior; The second possibility is the pressure and flow are upstream of the broken lattice, meaning that the mutant directly impacts pressure/flow that has consequences on lattice formation.

We favor the second scenario because the first scenario still leaves it unresolved how Scabrous can pattern differentiation by lateral inhibition alone. The prior claims about Scabrous, as the reviewer in point 2 below states, is that it is a negative or positive signal for cell differentiation. But a fundamental and essential pre-requisite for the inhibitory signal to generate a regular lattice pattern of differentiation is that cells do not move relative to one another. The mathematical models of Shraiman and Baarkai only work assuming cells are stationary in the zones where Scabrous would hypothetically signal. Their vertex models explicitly have stationary cells. We have definitively shown that such an assumption is false – cells move alot! Thus, for Scabrous to work as a differentiation signal, it would need to signal to very nearby cells, which then flow with some periodicity into position and differentiate. It would not work by classic reaction-diffusion but as a hybrid mechanism. Formally, such a mechanism is possible though complicated. We favor the simplest model, which also fits with all published data on Scabrous in the eye disc. For no one has demonstrated a molecular signal transduction pathway downstream of Scabrous that directly regulates transcription in the eye. All current models are based on genetic perturbations of scabrous in the eye disc followed by staining of fixed tissue for markers like atonal. However, cell movements can alter expression patterns, and so such indirect genetic tests do not rigorously demonstrate Scabrous directly regulating gene expression in cells of the eye disc.

However, we understand the reviewer’s concerns and we therefore have modified the text to tone down the claims and present the alternative models as just discussed to better align with the reviewer’s concerns.

2. The major argument for the causal role of movement is the analysis of the scabarous mutant phenotype. This mutant background is complicated since there are claims that Scabarous is either a positive or a negative non-autonomous signal for photoreceptor specification. In addition, the effects of scabarous mutant phenotype are opposite immediately at the morphogenetic front (where it leads to a reduction in photoreceptor clusters), and several rows away (where excess clusters are observed due to catastrophes in periodicity of lateral inhibition patterns).

The reviewer’s point about cluster number is not completely accurate. At the MF there are clusters of multiple cells well above the usual 5 in number – in Figure 4A you can see one cluster with 17 cells. That cluster has 4 R8 cells as identified by tracking. This type of megacluster phenotype at the MF is not unusual. It is because too many cells stop moving at the same time and bud from the MF. The periodic high-flow zones are lost and so cells can bud off when they normally would not. The megaclusters later break apart, in that non-R cells intercalate as cell dilation and flow occur. This would account for the reviewer’s point that rows away, the number of clusters goes up. Their statement that the number goes up due to catastrophe in lateral inhibition in those posterior rows is not supported by our data.

The authors observed a partial reduction in cell dilation in the scabarous mutant background. While alterations in cell migration in scabrous mutant discs were scored, the authors cannot determine if these alterations are the cause for the spacing disruption or a consequence of aberrant signaling. In short, since the scabarous data is incomplete and inconclusive, it would be better to present it as an analysis of a key mutant with the new understanding of cell movement in mind, and discuss the observations without using them to try and prove the model.

We discussed this issue in point 1 and have modified the manuscript in accordance.

However, we by no means claimed in the manuscript that our results “proved” a model. We subscribe to the aphorism “All models are wrong but some are useful” (G.Box 1976 J Am Stat Assoc 71, 791). The model we favor is the simplest and least likely to be wrong. But we agree that other models should be discussed and we have now done so.

3. It would be helpful to relate the constant speed of furrow progression (formation of a new photoreceptor row every ~2.5 hours, driven by Hh and Dpp signaling), to the maximal migration speed of the cell clusters that do not differentiate (1.2-1.5 microns/hr). Are these cell clusters progressing with the furrow at the same speed?

Yes they are moving at the same speed. We stated this on lines 197-199. We also stated that we saw new columns emerge every 2 – 2.5 hours (lines 138-140).

4. In discussing their data, the authors focus strongly on the mechanism by which a periodic pattern of ommatidia is generated, but their interpretations actually constitute a considerably broader understanding and revision of the established developmental program for the differentiating eye imaginal disc. This "global picture", with its complexities, should be summarized in an encompassing model, which points out the key elements of their proposal in a single schematic (posterior and anterior "pressure zones", various cell flow patterns outside MF, transient immobility of ommatidial clusters, flow of clusters within the MF and their periodicity, onset of atonal expression, etc), to provide readers with a guide for piecing together a rather complex story.

This is an excellent suggestion. We now provide such a picture in Figure 7A.

Perhaps it is possible to consider a hybrid model for cell spacing, combining signaling and cell movement. In this model, cell migration would differentially impinge on the signaling distance between cells, but signaling would still be the initial cue. For example, cell clusters with reduced migration become more susceptible to inhibition by the older cluster, while the faster migrating cells become less susceptible and thus more prone to differentiate when the next photoreceptor row forms. A similar spacing argument was made for the potential role of constriction at the morphogenetic furrow, allowing lateral inhibition to reach more cells.

This is also something we discussed in point 1. The model in which Scabrous is still a differentiation inhibitor but cell flows are part of the patterning process. We have added discussion of such a model and modified the manuscript to be more open about the possibility.